# SoC-DT: Standard-of-Care Aligned Digital Twins for Patient-Specific Tumor Dynamics

## Abstract

Accurate prediction of tumor trajectories under standard-of-care (SoC) therapies remains a major unmet need in oncology. This capability is essential for optimizing treatment planning and anticipating disease progression. Conventional reaction–diffusion models are limited in scope, as they fail to capture tumor dynamics under heterogeneous therapeutic paradigms. There is hence a critical need for computational frameworks that can realistically simulate SoC interventions while accounting for inter-patient variability in genomics, demographics, and treatment regimens. We introduce **Standard-of-Care Digital Twin (SoC-DT)**, a differentiable framework that unifies reaction–diffusion tumor growth models, discrete SoC interventions (surgery, chemotherapy, radiotherapy) along with genomic and demographic personalization to predict post-treatment tumor structure on imaging. An implicit-explicit exponential time-differencing solver, IMEX-SoC, is also proposed, which ensures stability, positivity, and scalability in SoC treatment situations. Evaluated on both synthetic data and real world glioma data, SoC-DT consistently outperforms classical PDE baselines and purely data-driven neural models in predicting tumor dynamics. By bridging mechanistic interpretability with modern differentiable solvers, SoC-DT establishes a principled foundation for patient-specific digital twins in oncology, enabling biologically consistent tumor dynamics estimation. Code will be made available upon acceptance.

## 1 Introduction

Cancer is one of the most deadly disease and remains a leading cause of death in the United States alone with an estimated 2,041,910 new cases and 618,120 deaths in 2025 (Siegel et al., 2025). Predicting the tumor growth under standard-of-care (SoC) therapies remains a key challenge in mathematical oncology. Most solid tumors including gliomas, breast cancers, etc., despite having well-established protocols combining surgery, chemotherapy/immunotherapy, and radiotherapy (Stupp et al., 2005; Group et al., 2011), exhibit highly heterogeneous responses: some patients relapse within months, while others remain stable for years (Verhaak et al., 2010; Parker et al., 2009). This variability highlights the need for models that capture patient-specific spatio-temporal dynamics and provide actionable forecasts for treatment planning.

Existing clinical tools are largely limited to population-level survival statistics or static risk prediction (Harrell & Levy, 2022; Van Calster et al., 2019) that fail to account for the non-linear dynamics of tumor growth and treatment response. Imaging biomarkers such as volumetric measurements or radiomics features provide insightful information (Aerts et al., 2014; Kickingereder et al., 2016) but cannot predict tumor structures under different clinical interventions. Similarly, survival models capture covariate–outcome associations but do not model the underlying biology of tumor progression (Katzman et al., 2018; Yanagisawa, 2023; Liu et al., 2024). As a result, clinicians lack principled computational surrogates that can continuously evolve with the patient, adapt to incoming data, and evaluate the consequences of different treatment decisions. With the advent of generative AI, several methods have been proposed to generate missing (Bhattacharya et al., 2024, 2025a) or post-baseline (multiple timepoint) medical images (Bhattacharya et al., 2025b; Liu et al., 2025) but a major limitation of these approaches is their static nature: they are restricted to generating the next timepoint image from the previous one and lack the ability to incorporate heterogeneous inputs. This creates a critical technical gap in translating current generative AI methods into true digital twin modeling.

The concept of a *Digital Twin* has recently emerged as a paradigm for simulation-based modeling (Grieves, 2023; Laubenbacher et al., 2024; Venkatesh et al., 2022). A digital twin is a computational surrogate that evolves in tandem with its real-world counterpart, continuously adjusted via incoming observations. In oncology, this translates into patient-specific simulators that can forecast tumor trajectories under observed or hypothetical interventions, while assimilating multimodal evidence such as serial imaging, genomic profiles, and treatment logs (Björnsson et al., 2019; Corral-Acero et al., 2020; Kuang et al., 2024). From a machine learning perspective, digital twins align naturally with simulation-based approaches, as they require modeling tumor growth and existing treatment strategies. Their main purpose is to estimate potential clinical outcomes under different treatment choices. Despite these advantages, applying digital twins in oncology remains challenging due to limited longitudinal data, diverse patient populations, and disruptions introduced by interventions such as surgery, fractionated radiotherapy, and evolving treatment guidelines (Bruynseels et al., 2018; Katsoulakis et al., 2024; Yankeelov et al., 2015; Jarrett et al., 2018).

A natural foundation for oncology digital twins is provided by partial differential equations (PDEs) that model tumor growth from imaging scans. Classical reaction–diffusion models describe the spatio-temporal dynamics of tumor cell density as the interplay between local proliferation and spatial invasion, parameterized by biologically interpretable quantities such as net proliferation rate and diffusivity (Murray, 2007; Swanson et al., 2000; Unkelbach et al., 2014; Tracqui et al., 1995). Therapy effects can be incorporated via additional terms, for example chemotherapy-induced cytotoxicity or radiotherapy-induced survival fractions derived from linear and quadratic formulations (Fowler, 1989; Powathil et al., 2013). From the ML perspective, PDEs serve as powerful *structured priors*: they encode inductive biases about smoothness, conservation, and positivity that constrain the solution space and improve generalization under limited data. Recent advances in differentiable physics and neural PDE solvers (Raissi et al., 2019; Chen et al., 2018; Li et al., 2020; Brandstetter et al., 2022; Kovachki et al., 2023; Lu et al., 2021; Ruthotto & Haber, 2020) have demonstrated how mechanistic laws can be embedded directly into learning pipelines, enabling gradient-based calibration, hybrid data–physics models, and operator learning for spatiotemporal systems (Gupta & Brandstetter, 2022; Kovachki et al., 2023). This view reframes PDE-based oncology models not as standalone mathematical constructs but as differentiable operators that can be composed with deep neural architectures, calibrated end-to-end from multimodal data. Yet, practical deployment remains difficult: estimating parameters from sparse, noisy observations is ill-posed, genomic heterogeneity is rarely integrated into dynamical laws and existing methods often fail to handle the discontinuities induced by discrete interventions such as surgery, fractionated radiotherapy (Rockne et al., 2015) and different combinations and doses of chemotherapy and immunotherapy. Also, lack of multiple timepoint clinical data (such as images, genomic markers, demographics, etc.) to train a digital twin model has significantly restricted research in this domain.

To summarize, there is a lack for a framework that replicates the standard-of-care treatment methods and a tumor growth modeling method under different treatments. In this work, we propose the *first* **Standard-of-Care Digital Twin (SoC-DT)**, a framework that unifies tumor growth modeling and standard-of-care treatment methods. SoC-DT extends hybrid reaction–diffusion PDEs to incorporate the three pillars of SoC therapy, while integrating parameters like genomic markers (e.g., IDH1, MGMT, 1p/19q, ATRX, HER2, etc.). To render the model trainable and scalable, we introduce two innovations: (i) an IMEX–ETD solver that integrates diffusion implicitly and reaction/therapy dynamics analytically, ensuring stability, positivity, and efficiency on imaging ; and (ii) an event-aware adjoint method that propagates exact gradients through discontinuities introduced by surgery and radiotherapy, enabling gradient-based calibration from multimodal patient data. Since, there is no existing standardized dataset or framework, we propose a mechanism to generate synthetic datasets based on -standard-of-care treatments for digital twin experimentations. Together, these components yield a differentiable, biologically grounded model capable of personalized forecasts.

Our contributions are threefold: (i) **Standard-of-Care Digital Twins.** We formulate a unified PDE that couples tumor growth with SoC therapies and modulates key parameters via genomic and demographics markers. (ii) **Differentiable solvers.** We develop an IMEX–SoC, a standard-of-care based integration scheme and an event-aware adjoint method that ensure stability, positivity, and exact gradient propagation through discontinuities. (iii) **Comprehensive evaluation.** We demonstrate, on three synthetic datasets and a real longitudinal MRI dataset with genomic annotations, that SoC-DT outperforms both classical PDE baselines and black-box neural models in predicting tumor

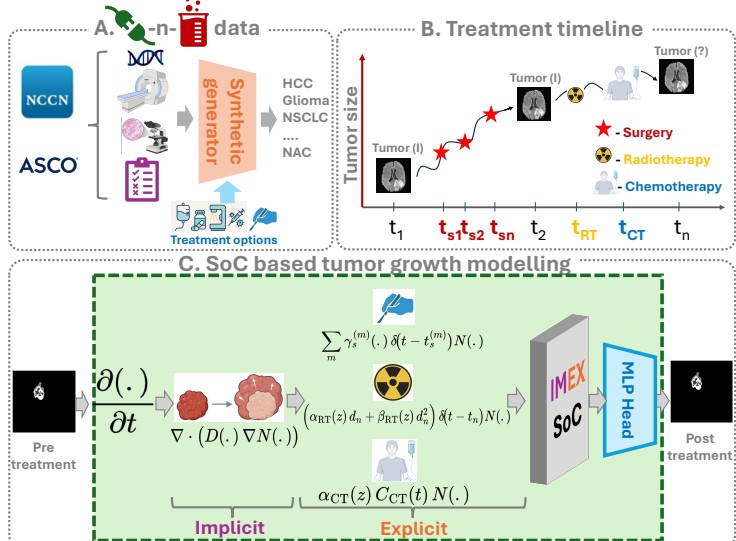

Figure 1: **Architecture**. A. A plug-and-play framework for generating synthetic datasets for different cancer types, B. An adaptation of a timeline for standard-of-care cancer treatment, C. Proposed *Standard-of-Care* tumor growth modeling framework. Post-treatment tumor structure is predicted from pre-treatment scans using a PDE framework that incorporates basic diffusion and proliferation terms, along with modules simulating surgery, chemotherapy, and radiotherapy.

evolution, treatment response, and survival. (iv) **Plug-n-Play framework.** We propose a framework for plug-and-play standard-of-care digital twin framework.

## 2 METHODS

### 2.1 BACKGROUND

**Reaction–diffusion models for tumor growth modeling.** Tumor cells grow in an unrestricted manner that involves diffusion and cell proliferation (Swanson et al., 2011). A reaction-diffusion model simulates this temporal evolution of the tumor cell density $N$, shown in the equation: $\frac{\partial N}{\partial t} = D\nabla^2 N + k\,N\left(1 - \frac{N}{\theta}\right)$, where $D > 0$ is the diffusion coefficient, $k > 0$ is the proliferation rate, and $\theta > 0$ is the carrying capacity. This reaction-diffusion model captures the dual processes of invasion (via diffusion) and logistic growth (via proliferation).

**Semi-discretization.** In the previous point, we discussed that reaction-diffusion PDEs models the evolution of tumor cell density, which in our case, is captured from imaging scans. In medical images, we discretize the 2D/3D spatial domain into $M$ pixels/voxels, yielding a vector $\mathbf{N}(t) \in \mathbb{R}^M$ of pixelwise/voxelwise tumor densities across time $t \in [0, T]$. Semi-discretization is one of the most standard ways to handle PDEs numerically by converting it to a system of ODEs in time.

$$\frac{d\mathbf{N}}{dt} = D\mathbf{L}\mathbf{N} + k\,\mathbf{N} \odot \left(1 - \frac{\mathbf{N}}{\theta}\right) - \alpha_{\mathrm{CT}}\,C(t)\,\mathbf{N}, \tag{1}$$

where $\mathbf{L}$ is the discrete Laplacian and $\odot$ denotes the Hadamard product. Treatment events such as surgery, radiotherapy (RT) and chemotherapy (CT) are represented as instantaneous multiplicative updates at intervention timepoints: $\mathbf{N}^+ = J(\mathbf{N}^-, \vartheta)$, $\vartheta = \{D, k, \alpha_{\mathrm{CT}}, \alpha_{\mathrm{RT}}, \beta_{\mathrm{RT}}\}$. Here, $N^+$ denotes the tumor state immediately before a treatment event, $N^-$ denotes the state immediately after the event and $J$ is the jump operator that maps the pre-event state to the post-event state. And, $\alpha_{\mathrm{RT}}$ and $\beta_{\mathrm{RT}}$ are radiosensitivity coefficients, and $\alpha_{\mathrm{CT}}$ is the chemo-sensitivity (cytotoxic) coefficient.

**Numerical solvers.** Explicit finite-difference schemes for diffusion are limited by the stability condition $\Delta t \leq \Delta x^2/(2D)$. In medical images, $\Delta x$ is small and $D$ can be large which makes $\Delta t$ to be extremely small (i.e. very slow simulations). To address this prior works have used implicit–explicit

exponential time-differencing (IMEX-ETD) schemes (Ascher et al., 1995) in which diffusion is treated implicitly using a conjugate gradient (CG) solve of $(I - \Delta t\, D\mathbf{L})\tilde{\mathbf{N}} = \mathbf{N}^n$. In addition to this, proliferation and chemotherapy are represented through a Riccati solution, $N_i^{n+1} = \frac{a\,\tilde{N}_i\,e^{a\Delta t}}{b\,\tilde{N}_i(e^{a\Delta t}-1)+a}$, where $a = k - \alpha_{\mathrm{CT}}C(t)$, $b = k/\theta$ with bounds $0 \le N \le \theta$.

## 2.2 FRAMEWORK

Designing a digital twin framework is essential for systematically modeling patient-specific disease trajectories and treatment responses. Our proposed framework is organized into two stages: *(a) Plug-and-Play dataset generation*, which ensures flexible and standardized data preparation, and *(b) Treatment timeline*, which captures the sequential flow of clinical interventions and imaging follow-up.

**Plug-and-Play dataset generation.** We propose a novel method for generating synthetic treatment scenarios to facilitate digital twin modeling. We synthesize the following for multiple cancer types: phantom images (either MRI scans or CT images) at different timepoints (i.e. pre-treatment, post-surgery and post-treatment), the genomic and molecular markers (such as IDH1 for AG, HER2 for breast, etc.), demographics (such as Age, Gender etc.), time-to-treatment and standard-of-care treatment (such as radiotherapy, chemotherapy, surgery, etc.). The standard-of-care for different tumor types is obtained from different public guidelines namely NCCN and ASCO. These guidelines provide detailed standards for different treatments under different genomic markers, histology, grade, etc. For example, for a post-menopausal patient with Ductal or Lobular histology and with HR status ER positive and or PR positive and HER2 negative, if pN0, the suggested therapy is adjuvant endocrine therapy and if pN2/pN3, the suggested therapy is adjuvant chemotherapy followed by endocrine therapy (More details in Appendix A).

**Treatment timeline.** Following this, we adapt a generalized timeline from clinically-defined standard-of-care treatment methods (Baden et al., 2024; Gradishar et al., 2024). Most solid tumor treatments follow a structured sequence that begins with pre-treatment imaging to establish the diagnosis and staging, often accompanied by a biopsy for histopathology and molecular profiling. This is followed by a primary intervention, typically surgery if feasible, or neoadjuvant therapy when tumors are inoperable. Post-operative imaging is then obtained to evaluate the extent of resection and provide a new baseline for subsequent care. Adjuvant therapies, including radiation, chemotherapy, targeted agents, or immunotherapy, are administered depending on tumor type and biology. Finally, post-treatment imaging is performed to assess therapeutic response and detect residual disease, after which patients transition to regular surveillance. This general timeline may be adapted with neoadjuvant therapy before surgery, non-surgical definitive treatment, enrollment in clinical trials, or palliative approaches in advanced disease.

## 2.3 STANDARD-OF-CARE DIGITAL TWINS

Following this framework, we develop an end-to-end digital twin for tumor growth modeling. A vanilla reaction–diffusion model, however, cannot incorporate the complexities of standard-of-care treatments and is therefore limited in its ability to accurately capture patient-specific growth dynamics. To address this, we formulate a *Standard-of-Care Digital Twin (SoC-DT)* as a patient-specific biophysical rollout on the imaging (MRI in our case) domain $\Omega \subset \mathbb{R}^D$:

$$\frac{\partial N(x,t)}{\partial t} = \underbrace{\nabla \cdot \big(D(z)\,\nabla N(x,t)\big)}_{\text{Diffusion/invasion}} + \underbrace{k(z)\,N(x,t)\Big(1 - \tfrac{N(x,t)}{\theta}\Big)}_{\text{Logistic proliferation}} - \underbrace{\alpha_{\mathrm{CT}}(z)\,C_{\mathrm{CT}}(t)\,N(x,t)}_{\text{Chemotherapy kill}}$$

$$- \underbrace{\sum_m \gamma_s^{(m)}(x)\,\delta\big(t - t_s^{(m)}\big)N(x,t)}_{\text{Surgery resection}} - \underbrace{\sum_n \Big(\alpha_{\mathrm{RT}}(z)\,d_n + \beta_{\mathrm{RT}}(z)\,d_n^2\Big)\delta\big(t - t_n\big)N(x,t)}_{\text{Radiotherapy}}.$$

$$(2)$$

Here, $z$ denotes patient-specific covariates (e.g., demographics, grade, molecular markers) that modulate biophysical parameters; $\gamma_s^{(m)}(x)$ is the spatial resection mask for the $m$-th surgery, indicating the fraction of tumor removed at pixel/voxel $x$; $m$ indexes surgery events with times $t_s^{(m)}$; $\delta(\cdot)$ is the Dirac delta distribution enforcing instantaneous treatment events; $\alpha_{\mathrm{RT}}(z)$ and $\beta_{\mathrm{RT}}(z)$ are radiotherapy parameters conditioned on patient-specific covariates $z$ (linear and quadratic terms of the

linear–quadratic model, respectively); $d_n$ is the radiation dose per fraction delivered at time $t_n$; $t_s^{(m)}$ and $t_n$ denote the calendar days to surgery and radiotherapy fractions, respectively.

## 2.4 SPATIAL DISCRETIZATION

We numerically solve the SoC-DT directly on the native image lattice of the medical images, treating each pixel/voxel as a computational grid point. The Laplacian operator $\mathbf{L}$ acts on the tumor field $\mathbf{N}$ to approximate its second spatial derivatives, with reflective (Neumann) boundary conditions imposed to prevent artificial flux of tumor tissues across the brain boundary. Note that, $\mathbf{N}$ is the vector of tumor tissue densities obtained from $N$ which is a function of space ($x$) and time ($t$). This semi-discretized form is expressed as a system of ordinary differential equations in time:

$$\frac{d\mathbf{N}}{dt} = D\,\mathbf{L}\,\mathbf{N} + k\,\mathbf{N} \odot \left(1 - \tfrac{\mathbf{N}}{\theta}\right) - \alpha_{\mathrm{CT}}\,\mathbf{C}(t) \odot \mathbf{N} - \sum_m \mathbf{R}^{(m)} \odot \mathbf{N}\,\delta\big(t - t_s^{(m)}\big)$$
$$- \sum_n \big(\alpha_{\mathrm{RT}}d_n + \beta_{\mathrm{RT}}d_n^2\big)\,\mathbf{N}\,\delta\big(t - t_n\big), \tag{3}$$

where $\mathbf{R}^{(m)} \in [0,1]^{|\Omega|}$ is the voxel-wise resection fraction for the $m$-th surgery and jump conditions for RT at interval ends. This ensures numerical stability and efficiency by avoiding mesh generation or resampling.

## 2.5 IMEX–SOC SOLVER

A key challenge of PDE-based digital twin is that the tumor growth dynamics is stiff Cristini & Lowengrub (2010). Diffusion is numerically unstable under explicit schemes and on the other hand non-linear proliferation and treatment events are difficult to integrate implicitly at scale. To address this, we propose a novel implicit–explicit *Standard-of-Care* (IMEX–SoC) solver, which is a hybrid exponential time-differencing (ETD) scheme that ensures stability, efficiency, and proper handling of discontinuous treatment events. To balance stiffness and efficiency we propose these three steps: **Implicit diffusion:** The update term $(\mathbb{I} - \Delta t\,D\,\mathbf{L})\tilde{\mathbf{N}} = \mathbf{N}^n$ propagates the diffusion effect using an implicit finite-difference step. Unlike explicit schemes, where $N^{n+1}$ is computed directly from $N^n$, which makes stiff problems like diffusion unstable, unless the time step $\Delta t$ is extremely small. **Closed-form reaction/chemo:** This step solves the local ODE, which combines logistic proliferation with chemotherapy kill. Its closed-form Riccati solution is $N^{n+1} = \frac{a\,\tilde{N}\,e^{a\Delta t}}{b\,\tilde{N}(e^{a\Delta t}-1)+a}$, $a = k - \alpha_{\mathrm{CT}}C$, $b = k/\theta$, clamped to $[0, \theta]$ to preserve biologically valid tumor densities. **Treatment events:** In this step, we discuss how tumor density is updated by instantaneous jump conditions. Surgery resection is modeled as $N^+ = (1 - R) \odot N^-$, where $R(x)$ is the pixel/voxel-wise resection mask. And for radiotherapy, tissue survival rate follows a linear–quadratic model represented as $N^+ = S(d) \odot N^-$, $S(d) = \exp(-\alpha_{\mathrm{RT}}d - \beta_{\mathrm{RT}}d^2)$, with $d$ the dose per fraction parameterized through radiosensitivity parameters $\alpha_{\mathrm{RT}}$ and $\beta_{\mathrm{RT}}$. *Time handling:* To reduce artificial uniform time spacing and preserve real calendar-day gaps between scans, we simulate each interval of $\Delta t$ days with $n = \lceil \Delta t \times \mathrm{steps\_per\_day} \rceil$ sub-steps, ensuring alignment between solver trajectories and actual clinical follow-up times.

## 2.6 TRAINING

To stabilize trajectories, we incorporate an optional assimilation step at intermediate observed time-points, where predictions $u$ are blended with ground-truth masks $u_{\mathrm{obs}}$ as $u \leftarrow \alpha u + (1 - \alpha)u_{\mathrm{obs}}$, $\alpha \in [0,1]$. For training, we use a Dice loss function which minimizes the DSC score between predicted and ground-truth post-treatment tumor masks. We use the Adam optimizer with gradient clipping to ensure stable convergence. Therotical gurarantees in Appendix D.

## 3 EXPERIMENTS AND RESULTS

### 3.1 EXPERIMENTAL SETUP

**Datasets.** For quantitative comparisons and sensitivity analysis, we generated three longitudinal synthetic datasets namely AG (Brain Cancer), HCC (Liver Cancer) and NAC (Breast Cancer). For

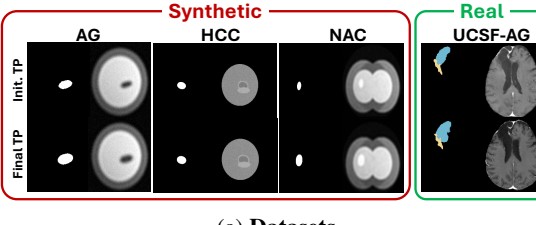

(a) **Datasets.**

| Method | MAE ($\downarrow$) | RMSE ($\downarrow$) |
|---|---|---|
| Linear | $210.4 \pm 108.5$ | $293.3 \pm 147.2$ |
| Fisher | $214.2 \pm 99.5$ | $307.5 \pm 130.5$ |
| Hybrid | $218.7 \pm 103.2$ | $309.4 \pm 142.8$ |
| PINN | $214.2 \pm 99.5$ | $307.5 \pm 130.5$ |
| Ours | $\mathbf{200.4 \pm 112.5}$ | $\mathbf{279.5 \pm 154.6}$ |

(b) **Progression analysis.**

Figure 2: (a) For our experiments, we use 3 synthetic datasets and 1 real clinical dataset. Initial and final timepoint images along with the different treatment methods are shown. AG: Adult Glioma, HCC: Hepatocellular Carcinoma, and NAC: Neoadjuvant Chemotherapy for Breast cancer. We perform quantitave analysis on both synthetic and real datasets. Synthetic datasets are primarily used for stress testing and real datasets are used for clinical downstream tasks (b) Comparison of our method against baseline models for regression of time-to-progression (days), evaluated using MAE and RMSE ($\mu \pm \sigma$ is reported).

Table 1: Comparison of DSC results across datasets and model variants. DSC ($\uparrow$) higher is better.

(a) **Quantitative comparisons**

| | Synthetic | | | Real |
|---|---|---|---|---|
| | **AG** | **HCC** | **NAC** | **UCSF** |
| **UNet** | $34.05 \pm 45.49$ | $44.50 \pm 9.08$ | $4.50 \pm 4.47$ | $53.09 \pm 2.96$ |
| **ConvLSTM** | $66.10 \pm 37.08$ | $41.99 \pm 23.66$ | $7.50 \pm 2.50$ | $37.69 \pm 31.19$ |
| **Linear PDE** | $75.52 \pm 1.79$ | $48.20 \pm 8.20$ | $81.28 \pm 2.03$ | $52.78 \pm 3.49$ |
| **Fisher PDE** | $75.64 \pm 1.79$ | $48.20 \pm 8.20$ | $76.08 \pm 2.32$ | $52.75 \pm 3.47$ |
| **PINN** | $75.57 \pm 1.73$ | $48.30 \pm 8.18$ | $78.62 \pm 3.02$ | $51.71 \pm 4.61$ |
| **Hybrid PDE** | $77.01 \pm 1.99$ | $48.76 \pm 8.27$ | $84.41 \pm 2.39$ | $54.29 \pm 3.14$ |
| **Ours** | $\mathbf{85.26 \pm 1.69^*}$ | $\mathbf{50.46 \pm 8.45}$ | $\mathbf{86.20 \pm 3.08^*}$ | $\mathbf{55.45 \pm 8.12}$ |

(b) **Ablation analysis**

| Heads / Terms | DSC($\mu \pm \sigma$) |
|---|---|
| Patient-Emb. | $20.31 \pm 1.434$ |
| MLP | $82.39 \pm 2.179$ |
| Linear | $82.39 \pm 2.179$ |
| Global | $82.39 \pm 2.179$ |
| No-growth | $75.50 \pm 1.759$ |
| No-diff | $81.94 \pm 2.376$ |
| No-kill | $82.66 \pm 2.269$ |
| No-surg | $82.66 \pm 2.269$ |

clinical applications, we use a real adult longitudinal post-treatment diffuse glioma dataset, UCSF-ALPTGD Fields et al. (2024) (Figure 2 a), Details in Appendix A.

**Baselines.** To comprehensively evaluate our proposed framework, we benchmarked against a diverse set of baselines spanning deep learning models, classical PDE-based methods, and hybrid physics–informed approaches. Specifically, we considered six representative methods grouped into three categories: Data-driven deep learning models: **UNet** (Ronneberger et al., 2015) and **ConvLSTM** (Shi et al., 2015), which learn spatiotemporal tumor evolution directly from data without explicit physics constraints. Classical PDE-based models: **Linear PDE** and **Fisher KPP** (Fisher, 1937; Kolmogorov, 1937), which represent traditional reaction–diffusion formulations commonly used to model tumor growth dynamics. Physics-informed NNs: **PINN** (Raissi et al., 2019) and **Hybrid PDE** (Sun et al., 2020), which integrate PDE priors with neural correctors (Hybrid).

**Evaluation metrics.** To evaluate the tumor growth modeling, we compute the Dice–Sørensen Coefficient (DSC) scores of the predicted masks with ground truth masks. For progression analysis experiments, we compute Mean Absolute Error (MAE) and Root Mean Squared Error (RMSE). All experiments are performed in a 5-fold cross validation setting; we report the mean ($\mu$) and standard deviation ($\sigma$) on 5 folds.

## 3.2 CLINICAL APPLICATIONS

For clinical applications, we evaluate for post-treatment tumor mask prediction and time-to-progression prediction.

**Setup.** Global fold-level scalars $(D, k, \alpha_{\mathrm{CT}}, \alpha_{\mathrm{RT}}, \beta_{\mathrm{RT}})$ and modulates $(D, k, \alpha_{\mathrm{CT}})$ per patient are learned via a genomic MLP with MGMT, IDH, 1p/19q, grade, and age as inputs. Standard-of-care events are explicitly modeled through exponentially decaying *chemo* pulses derived from first-chemo timing/type and a *radiotherapy* schedule of $30 \times 2$ Gy fractions if RT begins after baseline. Training follows patient-wise K-fold cross-validation with Adam, gradient clipping, and a mixed Dice+BCE loss; SOC-DT conditions on baseline and first post-treatment masks and rolls out to predict the second follow-up, with assimilation at pre-final timepoints.

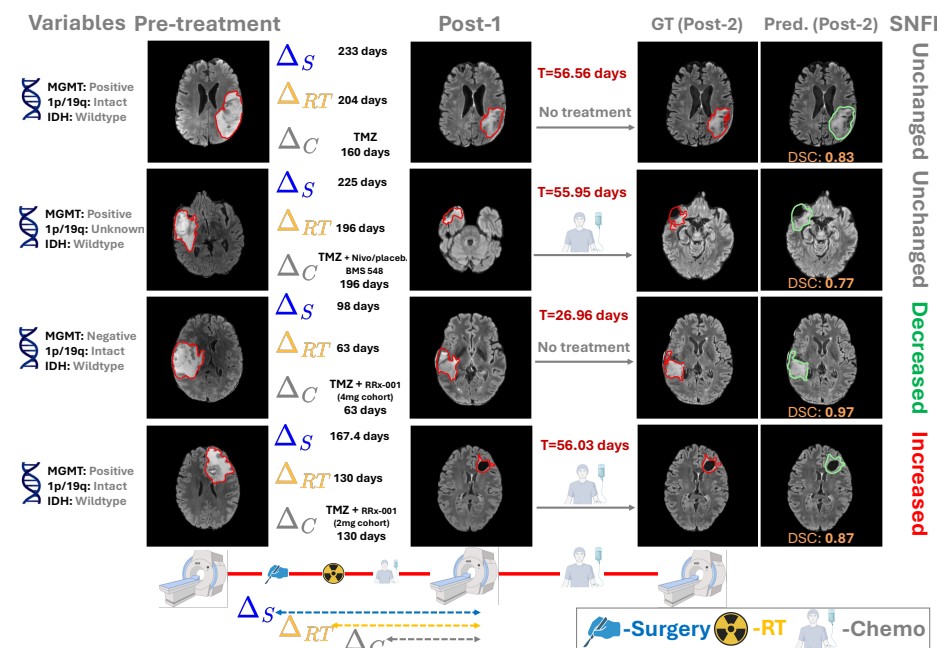

Figure 3: **Standard-of-Care Digital Twins.** Examples of SoC-DT projections comparing predicted post-treatment tumor masks with the ground-truth follow-up tumor masks, given pre-treatment and post-operative tumor masks overlaid on the corresponding FLAIR sequences. Different treatment options (surgery, radiotherapy, chemotherapy) are illustrated, with the elapsed time from post-operative to post-treatment scan indicated. Quantitative changes in SNFH volume are also reported.

**Post-treatment tumor estimation.** On the UCSF dataset, classical PDE models (Linear: $52.78 \pm 3.49$; Fisher: $52.75 \pm 3.47$) and physics-informed approaches (PINN: $51.71 \pm 4.61$; Hybrid PDE: $54.29 \pm 3.14$) achieved similar performance. Our method reached the highest DSC of $55.45 \pm 8.12$, representing a 1.16-point absolute gain over Hybrid PDE. This improvement demonstrates that incorporating standard-of-care (SoC) components—surgery, radiotherapy, and chemotherapy—into the governing equations yields measurable performance gains. By extending the Hybrid PDE framework with these clinically grounded factors, our approach better captures treatment-modulated tumor evolution and achieves superior generalization on real-world data.

**Time-to-progressions results.** Table 2.b summarizes the time-to-progression (TTP) prediction performance across different baselines. Among the classical physics-based models, the linear, Fisher, and hybrid formulations achieved mean absolute errors (MAE) of $210.4 \pm 108.5$, $214.2 \pm 99.5$, and $218.7 \pm 103.2$ days, respectively. The PINN baseline performed comparably to the Fisher model, reflecting limited benefit from the additional physics-informed constraint in this setting. Our proposed approach achieved the lowest MAE ($200.4 \pm 112.5$ days) and RMSE ($279.5 \pm 154.6$ days), demonstrating improved predictive accuracy over all baselines.

**Qualitative analysis.** We qualitatively assess the tumor response predicted by SoC-DT in comparison to PINN and Hybrid PDE. Figure 4.A illustrates representative post-treatment FLAIR sequences with the ground truth (GT) and predicted tumor masks overlaid. SoC-DT demonstrates improved DSC score and more accurately captures the residual enhancing margins and infiltrative components, while the baselines frequently overestimate tumor boundaries. Importantly, SoC-DT avoids overestimation of peritumoral edema, thereby maintaining closer agreement with radiologically defined boundaries. In Figure 3, we show examples of predicted tumor masks under varying treatment scenarios, alongside GT pre-treatment, post-treatment timepoint 1, and post-treatment timepoint 2 FLAIR scans with their corresponding annotations. SoC-DT produces masks that closely align with radiological findings, reflecting both the expected reduction in surrounding nonenhancing fluid-attenuated inversion recovery hyperintensity (SNFH) volume after surgery and the subsequent treatment-modulated changes. The predictions demonstrate preservation of anatomical context, with post-treatment contours that mirror GT evolution in size and morphology. These observations show the clinical utility of SoC-DT in simulating realistic treatment trajectories, capturing both the effi-

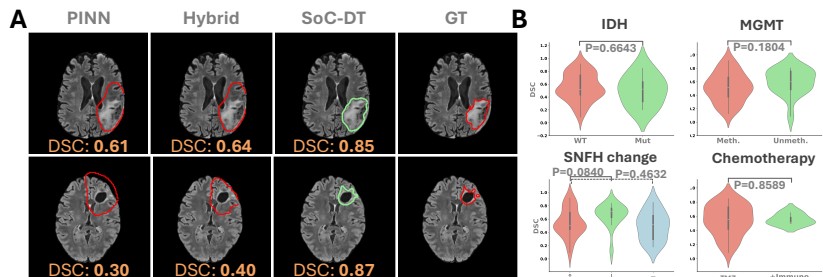

Figure 4: A. **Qualitative comparisons.** We compare the SoC-DT generated post-treatment tumor masks with baselines like PINN and Hybrid PDE. We also report the DSC scores, B. **Sensitivity analysis.** Box plots are shown for DSC scores across different genomic marker types, SNFH change and chemotherapy/+immunotherapy.

cacy and limitations of therapeutic interventions in a manner concordant with expert interpretation. **Sensitivity analysis.** Here, we show that SoC-DT is robust for different clinical parameters such as genomic markers, SNFH tumor volume change, and treatment types. In Figure 4 B, we show violin plots of DSC scores for different clinical parameters. We observe that fo all the different parameters there is no statistically significant difference in the DSC scores, demonstrating that SoC-DT is robust.

### 3.3 COMPARISON ON TOY DATASETS AND SENSITIVITY ANALYSIS

**Quantitative results.** We first evaluated SoC-DT on the three synthetic cohorts (AG, HCC, NAC) to benchmark performance under controlled tumor evolution and treatment settings. As shown in Table 1a, classical PDE models (Linear, Fisher) achieved DSC scores in the range of 75–81, capturing broad growth dynamics but without treatment-specific fidelity. Physics-informed approaches (PINN, Hybrid PDE) offered modest improvements, with Hybrid PDE reaching 77.01 on AG and 84.41 on NAC. By contrast, SoC-DT consistently achieved the highest performance across all synthetic datasets, with DSC improvements of +8.25 over Hybrid PDE on AG and +1.79 on NAC. Notably, the HCC cohort, which exhibits more heterogeneous kinetics, proved more challenging, with all methods performing in the mid-40s to low-50s; nonetheless, SoC-DT achieved the best score of 50.46, indicating improved generalization across diverse tumor growth regimes. These results highlight the robustness of incorporating SoC interventions directly into the model, yielding measurable gains across multiple synthetic scenarios.
**Sensitivity analysis.** We evaluated SoC-DT's robustness to key hyperparameters, including prediction horizon($\Delta t$), assimilation strength ($\alpha$), sub-steps per day ($N$), and carrying capacity ($K$). Detailed analysis is in Appendix C. To summarize the findings, SoC-DT showed stable performance across different parameters demonstrating robustness and clinical reliability.

### 3.4 ABLATION ANALYSIS

To evaluate the relative importance of architectural choices and mechanistic components, we conducted two sets of ablation experiments: one focused on prediction heads and physics modules, and another on the formulation of kill and surgical interventions (Table 1b and Appendix Table 3). **Model ablations:** Across different encoder heads, the MLP, linear, and global pooling designs achieved nearly identical performance (DSC $\approx 0.824$), suggesting that segmentation accuracy was not strongly dependent on the choice of simple aggregation mechanism. In contrast, replacing the head with a patient-embedding–based module caused a marked degradation (DSC $= 0.203 \pm 0.012$), indicating that identity-level embeddings alone were insufficient to capture tumor dynamics. We next ablated the mechanistic terms underlying tumor evolution. Removing the diffusion term led to only a modest decrease (DSC $= 0.819 \pm 0.022$), consistent with a cohort where volumetric growth dominated over spatial spread. By comparison, excluding the growth component substantially reduced performance (DSC $= 0.755 \pm 0.015$), underscoring its central role in shaping disease trajectories. In contrast, omitting either the kill or surgery terms yielded no measurable degradation (DSC $\approx 0.826$), implying that these effects were either sparsely represented in the dataset or partially re-

dundant with other modeled processes. Collectively, these ablations reveal that accurate forecasting relies most critically on explicit modeling of growth dynamics.

### 3.5 DISCUSSION

Our experiments demonstrate that SoC-DT effectively integrates standard-of-care interventions into a mechanistic tumor evolution framework, achieving superior predictive accuracy over both classical PDE models and existing physics-informed neural networks. The quantitative and qualitative results indicate that explicitly modeling surgery, radiotherapy, and chemotherapy improves mask fidelity. Sensitivity analyses further highlight the robustness of the approach to variations in prediction horizon, assimilation strength, temporal discretization, and carrying capacity, suggesting that the framework can generalize across diverse clinical scenarios without extensive hyperparameter tuning. Ablation studies emphasize the central role of growth dynamics, while architectural choices and auxiliary terms contribute more modestly, emphasizing that clinically grounded mechanistic modeling drives performance gains. Collectively, these findings suggest that SoC-DT provides a reliable and interpretable tool for simulating patient-specific tumor trajectories, bridging the gap between predictive modeling and clinically actionable insights.

## 4 RELATED WORK

**Mathematical oncology and reaction–diffusion models.** The use of partial differential equations (PDEs) to describe tumor growth has a long tradition in mathematical oncology. Early reaction–diffusion models captured the balance between local proliferation and invasion into surrounding tissue (Konukoglu et al., 2009; Mang et al., 2012; Murray, 1989; Swanson et al., 2000; 2002), with parameters such as the diffusion coefficient and proliferation rate estimated from imaging. Subsequent extensions incorporated treatment effects, including cytotoxic chemotherapy (Konukoglu et al., 2009; Stamatakos et al., 2010; Swanson et al., 2000) and radiotherapy via the linear–quadratic survival model (Rockne et al., 2010; Fowler, 1989). While these models provide interpretable parameters and mechanistic insights, they are rarely used clinically because they are difficult to calibrate robustly to patient-specific data, often assume homogeneous populations, and typically neglect the impact of discrete events such as surgery. Moreover, the computational burden of solving PDEs on high-resolution 3D MRI grids has limited their adoption in practice.

**Digital twins in oncology.** The concept of a digital twin in cancer has recently attracted attention as a means of building computational surrogates of patient trajectories. Prior works have used statistical models or machine learning to forecast tumor growth under observed therapy (Gerlee, 2013; Chaudhuri et al., 2023). However, most existing digital twin approaches either rely on population-level statistics or purely data-driven predictors, and few integrate mechanistic PDE models with patient-specific molecular markers and treatment schedules. Our work differs by proposing a unified framework that combines reaction–diffusion dynamics, surgery/chemo/radiotherapy events, genomic personalization, and differentiable solvers. By embedding an IMEX–ETD integrator and an event-aware adjoint into the digital twin, we provide both theoretical guarantees and practical scalability, bridging the gap between mathematical oncology and machine learning (Ruthotto & Haber, 2020; Hao et al., 2022; Du et al., 2020; Sanchez-Gonzalez et al., 2020; Krishnan et al., 2017; Willard et al., 2022).

## 5 CONCLUSION

In this work, we introduced the Standard-of-Care Digital Twin (SoC-DT), a differentiable framework that unifies mechanistic tumor growth modeling with modern deep learning. By extending reaction–diffusion PDEs to incorporate surgery, chemotherapy, radiotherapy, and genomic modulation, and by developing stable IMEX–ETD solvers with event-aware adjoints, SoC-DT enables patient-specific calibration from multimodal data and biologically consistent forecasts of tumor evolution. Our experiments across multi-institutional cohorts demonstrate that SoC-DT outperforms both classical PDE baselines and black-box neural models, while retaining interpretability and supporting counterfactual simulation of alternative therapies. These results establish SoC-DT as a principled foundation for oncology digital twins, bridging mathematical oncology and machine learning to enable adaptive, patient-specific treatment planning.

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
