APPENDIX

Here we provide additional details, results, theoretical proofs and limitations of the proposed SoC-DT framework.

## A    DATASETS

In this work, we proposed a plug-and-play framework to generate synthetic datasets for experimentation. We synthesize three datasets, namely Adult Glioma (AG) for brain cancer, Hepatocellular Carcinoma (HCC) for liver cancer and Neoadjuvant Chemotherapy (NAC) for breast cancer. The Synthetic generation pipeline (shown in Figure 1 A), which is a reactive diffusion model generates these datasets. Details of the parameters of the reaction-diffusion model are shown in Figure 8. Details of the generated datasets are discussed below:

**AG (Brain Cancer):** For adult glioma, we seed lobulated, infiltrative masks within a brain phantom and evolve tumor occupancy with spatially varying diffusion/proliferation maps, while SoC impulses (surgery, fractionated RT, weekly chemo) modulate daily dynamics; imaging volumes (T1C, T2W, T2F) are rendered with bias fields, k-space low-pass filtering, and Rician noise, and per-timepoint masks/images are saved as .png/.npy (shown in Figure 5) with metadata for kinetics, genomics, and schedule. Detailed molecular information of the generated patients are shown in Table 2.

**NAC (Breast Cancer):** Breast NAC cases are generated on a torso–breast phantom with FGT-biased, possibly multifocal seeds; voxelwise PK curves (simplified Tofts + AIF) drive multi-phase DCE signals (pre/early/mid/late), scanner "styles" add motion/ghosting/Gibbs artifacts (shown in Figure 7), and patient-level response phenotypes (Responder→Hyperprogressor) modulate cycle-wise kill and regrowth, yielding RECIST-like diameters, optional PK proxy maps, and outcomes (pCR, survival). Details of molecular information of the generated patients are shown in Table 2.

**HCC (Liver Cancer):** For HCC, liver, vessels, and tumor are modeled with arterial hyperenhancement and portal/delayed washout; TACE events cause devascularization and optional lipiodol-like deposits, while SBRT applies a spatial dose map and LQ-inspired kill, and CT images are synthesized per phase (arterial/portal/delayed) under soft-tissue windowing with bow-tie bias and Gaussian noise (shown in Figure 6). Detailed demographics are shown in Table 2.

Across datasets, scan calendars are fixed or randomly sampled over clinically plausible horizons, and masks derived from thresholding the evolving field with small stochasticity to reflect acquisition variability. Each patient has per-timepoint file paths and quantitative fields (e.g., tumor area, RECIST-like diameter), and a companion patient-level table records genomics/biomarkers, kinetic parameters, intervention logs, and derived outcomes (event flags, survival days, toxicity where applicable). Design choices emphasize controllable realism (bias fields, k-space filtering, ghosting, Gibbs, windowing) and heterogeneity (white/gray matter or FGT/liver modulation) while remaining dependency-light and reproducible via a fixed RNG seed.

Figure 5: **AG (Brain Cancer).** An example case from the generated AG data where we show different phantom MR sequences (T1C, T2F, and T2W) along with the tumor masks for multiple timepoint visits.

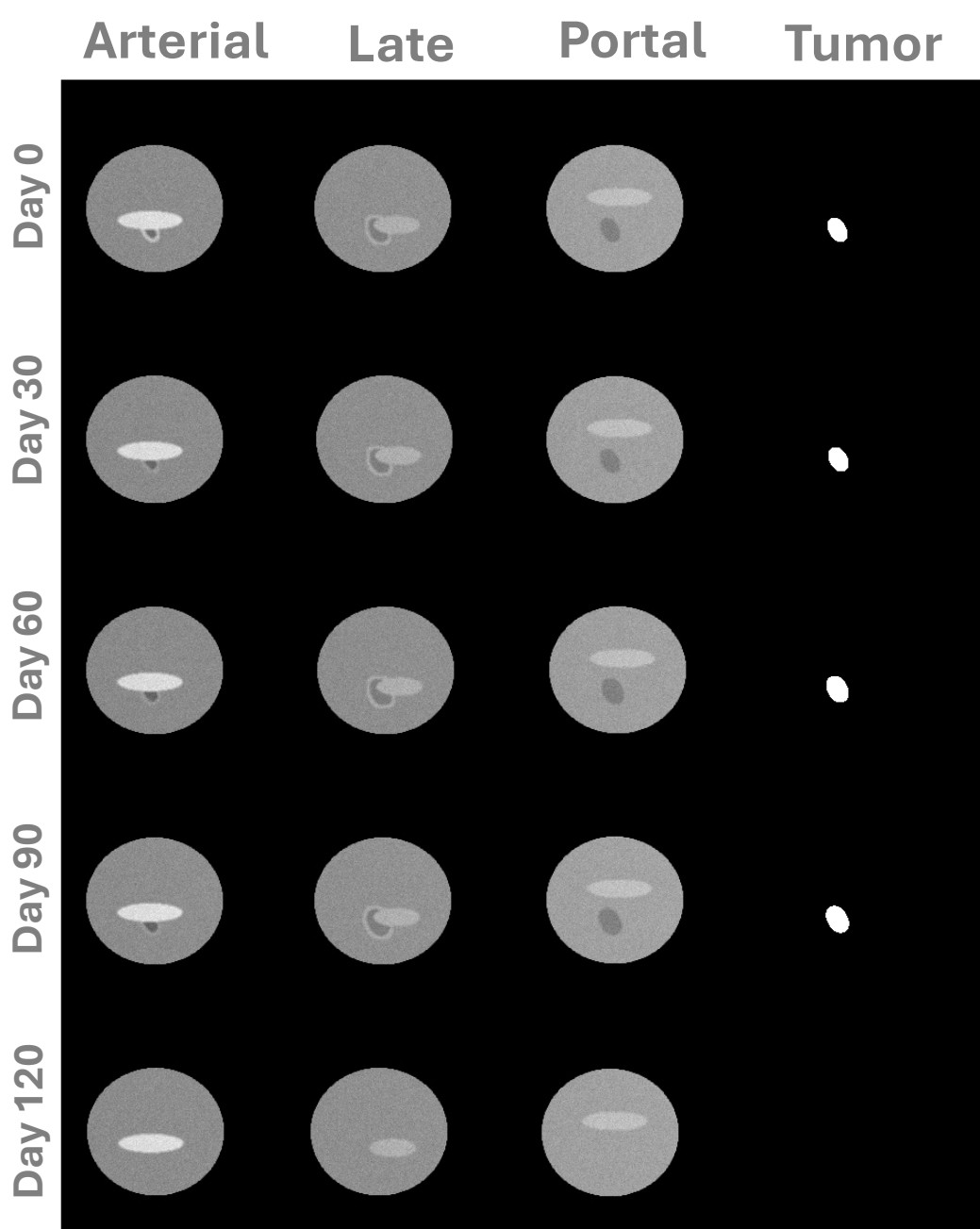

Figure 6: **HCC (Liver Cancer).** An example case from the generated HCC data where we show different phase enhanced CT images (Arterial, Portal and Late) along with the tumor masks for multiple timepoint visits.

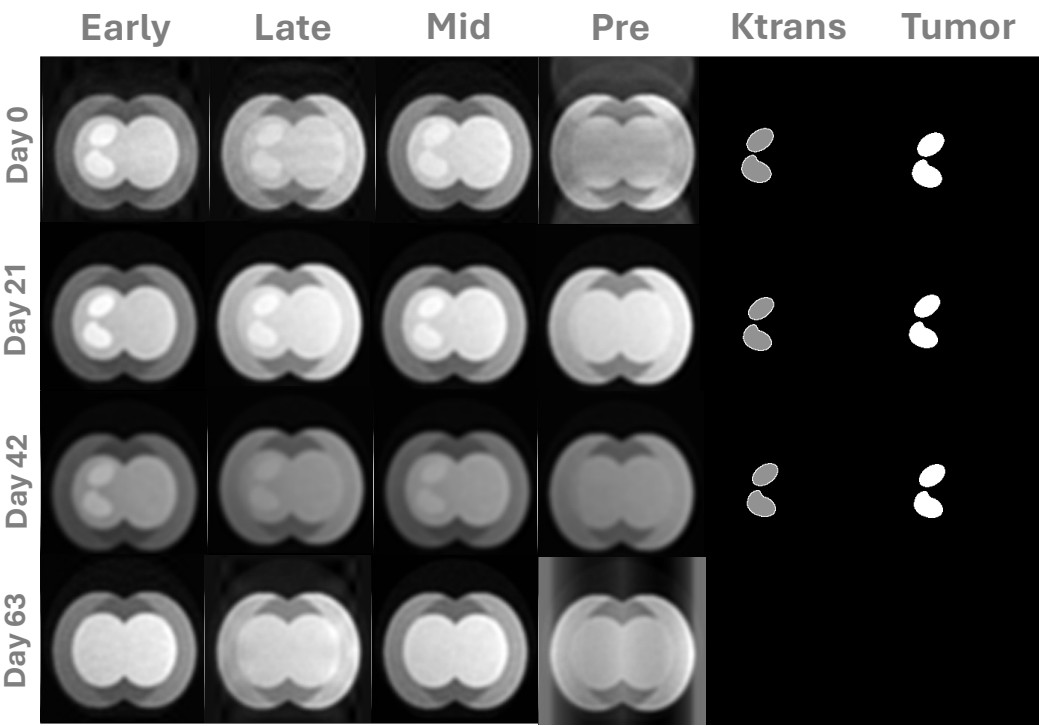

Figure 7: **NAC (Breast Cancer).** An example case from the generated NAC data where we show different phases DCE MRI along with the tumor masks and $K_{trans}$ image for multiple timepoint visits.

Table 2: Cohort characteristics and biomarker distributions across synthetic datasets: Adult Glioma (AG), Breast NAC, and Hepatocellular Carcinoma (HCC).

| Marker / Variable | Category | Count |
|---|---|---|
| **Adult Glioma (AG)** | | |
| IDH1 | Wildtype | 136 |
| | Mutant | 64 |
| MGMT | Unmethylated | 117 |
| | Methylated | 83 |
| EGFR | Normal | 132 |
| | Amplified | 68 |
| 1p/19q | Intact | 178 |
| | Codeleted | 22 |
| CDKN2A/B | Wildtype | 139 |
| | Deleted | 61 |
| TP53 | Wildtype | 111 |
| | Mutant | 89 |
| TERT | Mutant | 109 |
| | Wildtype | 91 |
| ATRX | Wildtype | 181 |
| | Loss | 19 |
| **Breast NAC** | | |
| ER | Positive | 138 |
| | Negative | 62 |
| PR | Positive | 129 |
| | Negative | 71 |
| HER2 | Negative | 149 |
| | Positive | 51 |
| **Hepatocellular Carcinoma (HCC)** | | |
| ALBI grade | 1 | 111 |
| | 2 | 77 |
| | 3 | 12 |
| BCLC stage | A | 97 |
| | B | 74 |
| | C | 29 |
| Event status | Death (1) | 135 |
| | Censored (0) | 65 |

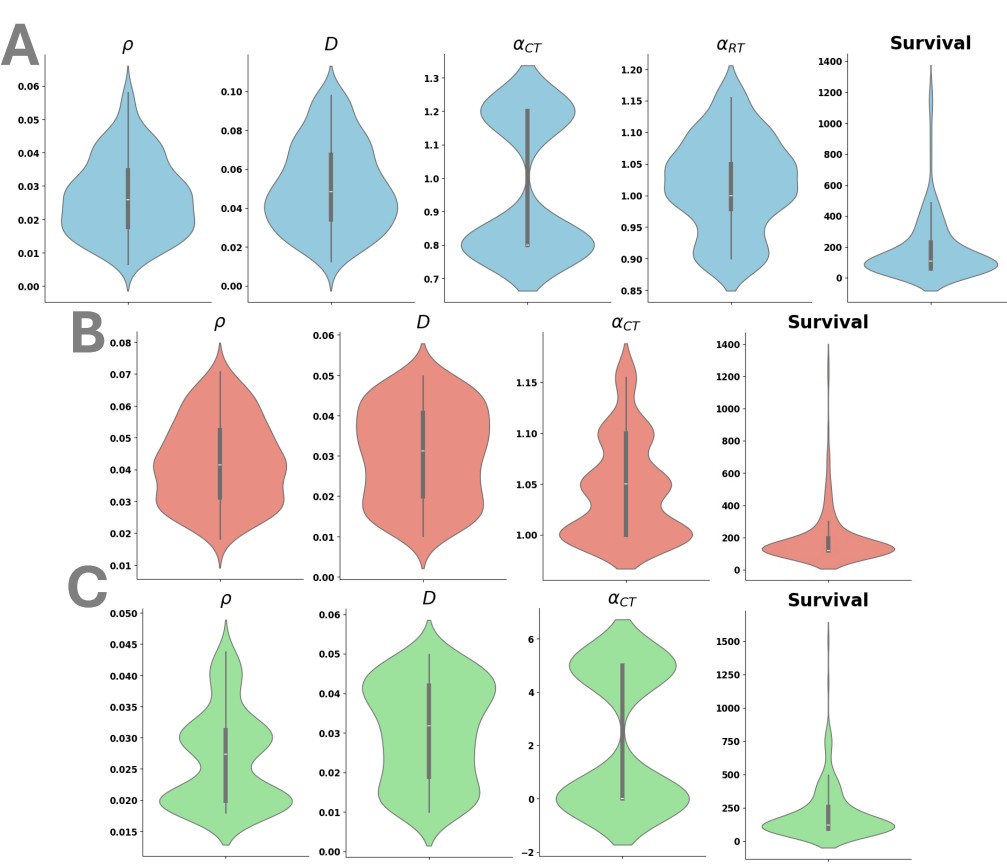

Figure 8: Box plots of different paramters of the Synthetic generator (shown in Figure 1 A) for datasets. AG hyperparameters are shown in A, HCC hyperparameters are shown in B and NAC hyperparameters are shown in C.

Table 3: Comparison of DSC results across Kill & Surgery settings @ $\tau = 0.5$.

| Setting | DSC($\mu \pm \sigma$) |
|---|---|
| Additive | $0.8457 \pm 0.0185$ |
| Saturation | $0.8468 \pm 0.0188$ |
| Synergy | $0.8466 \pm 0.0189$ |
| Mul | $0.8457 \pm 0.0185$ |
| Morph_op. | $0.8413 \pm 0.0243$ |
| Rim | $0.8450 \pm 0.0186$ |

## B    DETAILS OF KILL AND SURGERY SETTINGS

In this section, we provide further explanation of the different *kill* and *surgery* settings reported in Table 3.

### B.1    KILL SETTINGS

- **Additive:** The effect of cell kill is modeled as a direct additive reduction in the tumor population, proportional to treatment strength. This corresponds to a linear decrease in tumor density.
- **Saturation:** Introduces a nonlinear saturation term, where kill efficiency plateaus at higher doses, mimicking biological limits of treatment efficacy.
- **Synergy:** Models synergistic effects between treatments (e.g., chemotherapy + radiotherapy) using an interaction term, resulting in superlinear kill under combined interventions.

### B.2    SURGERY SETTINGS

- **Multiplicative:** A multiplicative reduction of tumor density in the resected region, representing partial tumor removal.
- **Morphological Operation:** A morphological erosion applied to the tumor mask, mimicking resection margins where adjacent tissue may also be removed.
- **Rim:** Explicit removal of a peripheral rim around the tumor, modeling surgical clearance margins commonly practiced to reduce recurrence risk.

These alternative formulations allow us to stress-test robustness of the model against different plausible implementations of treatment operators, while keeping the overall simulation framework consistent.

## C    ADDITIONAL RESULTS

Additional quantitative and qualitative results are shown here:

### C.1    QUANTITATIVE RESULTS

We perform the sensitivity analysis on toy datasets. In Figure 9, we report the results of our sensitivity analysis across numerical, biological, and evaluation parameters. Despite being exposed to a broad range of stress tests—including variations in numerical step size ($\Delta t$), treatment coupling ($\alpha$), post-processing thresholds ($\tau$), saturation level ($K$), and edema mask dilation—our model demonstrates a striking degree of robustness. Across almost all perturbations, the Dice similarity coefficient (DSC) remains tightly clustered in the 0.82–0.85 range, with standard deviations typically around 0.02–0.03, indicating stable generalization across folds. Even in cases where aggressive settings led to catastrophic HD95 outliers (e.g., very large $\Delta t$ or heavy dilation), the average DSC degraded only modestly, underscoring that the core tumor segmentation signal is preserved. Importantly, the model is least sensitive to the choice of $\tau$ and remains stable across data splits, highlighting resilience to

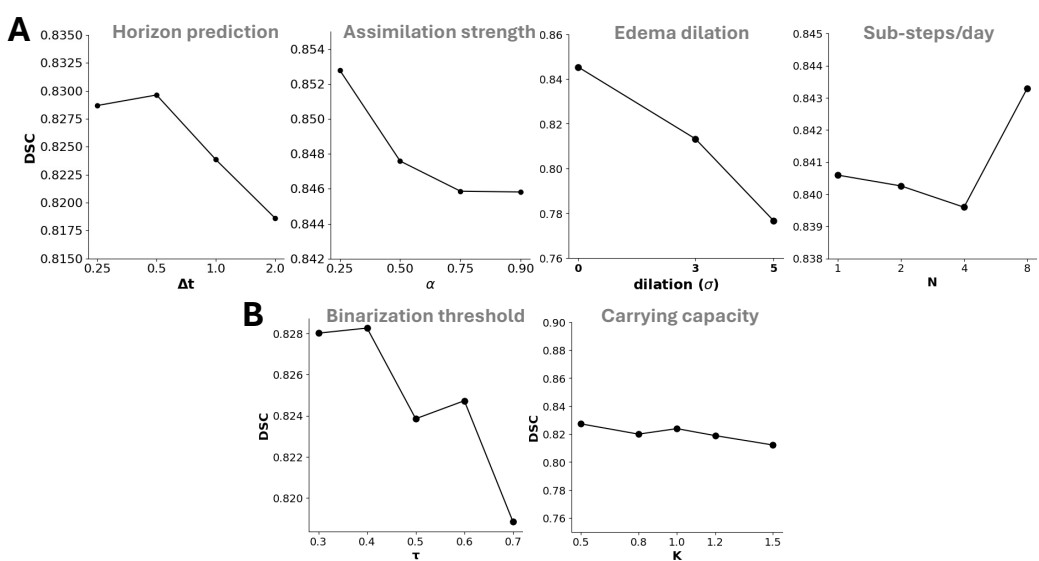

Figure 9: **Sensitivity analysis.**

evaluation criteria and sampling variation. The clearest sensitivities arise from preprocessing (edema dilation) and overly large numerical steps, yet even there the performance drop is gradual rather than abrupt. Taken together, these results suggest that our framework is robust under numerical, biological, and evaluation perturbations, maintaining consistently strong segmentation performance across a wide range of stress conditions.

## C.2 QUALITATIVE RESULTS

In Figure 10, we show additional results of our SoC-DT methods. And, we provide additional clarification regarding the timeline shown in Figure 3 and Figure 10. We map the UCSF PostopGlioma spreadsheet columns to our notation as follows: $\Delta_S$ = `Days from 1st surgery/DX to 1st scan`, $\Delta_{RT}$ = `Days from 1st scan to 1st RT start (neg = RT first)`, $\Delta_C$ = `Days from 1st chemo start to 1st scan`, and $T$ = `Days from 1st scan to 2nd scan`. The column `1st Chemo type` specifies the regimen associated with $\Delta_C$, while treatment status at the second scan is determined from `On tx at 2nd scan (c = chemo, rt, n = none)`, allowing classification into chemotherapy (`c`), radiotherapy (`rt`), both (`c+rt`), or no treatment (`n`).

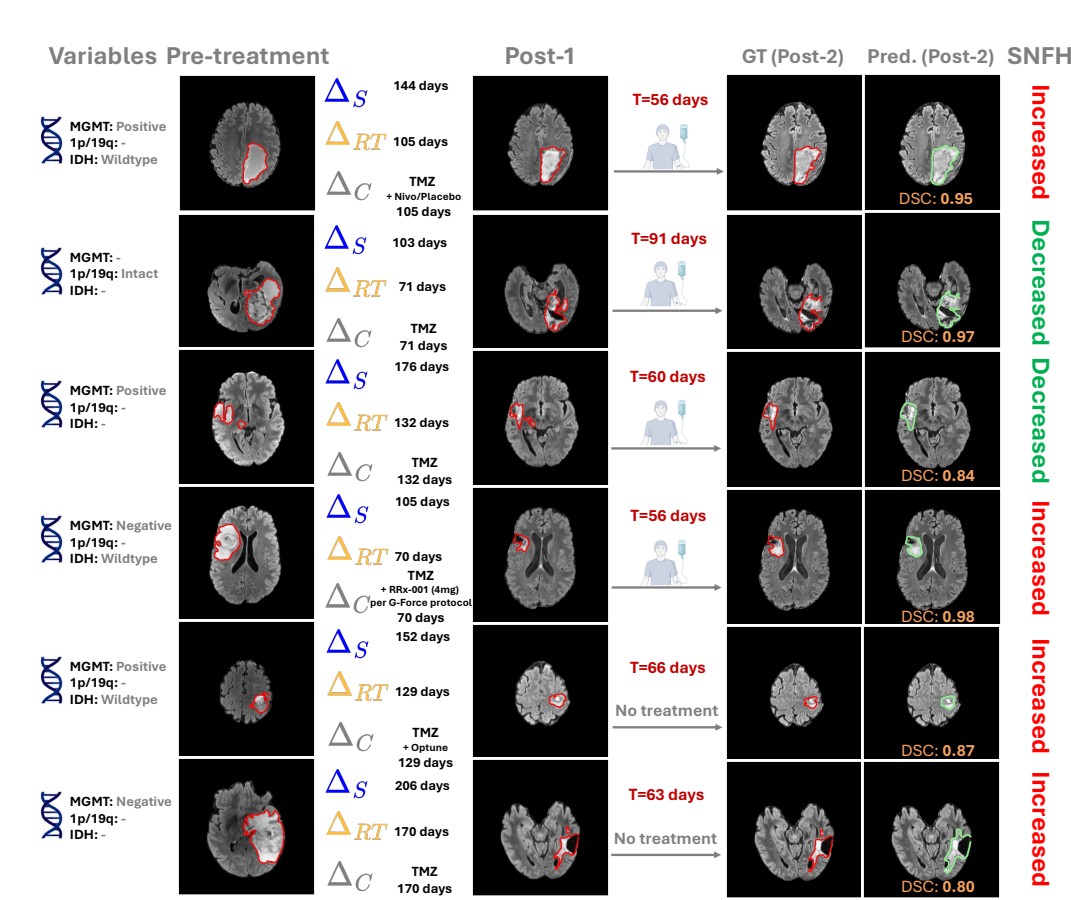

Figure 10: **Additional qualitative results.**

## D  THEORETICAL GUARANTEES

**Assumptions.** $\Omega \subset \mathbb{R}^d$ is a bounded Lipschitz domain with outward normal $\mathbf{n}$, Neumann boundary condition $\partial_{\mathbf{n}} N = 0$, constants $D, k, \theta > 0$, $\alpha_{\mathrm{CT}}, \alpha_{\mathrm{RT}}, \beta_{\mathrm{RT}} \geq 0$, chemo exposure $C_{\mathrm{CT}}(t)$ bounded and piecewise $C^1$ with finitely many jumps, and initial data $N_0 \in L^\infty(\Omega)$ with $0 \leq N_0 \leq \theta$ a.e. Event times $\mathcal{E} = \{t_s^{(m)}\} \cup \{t_n\}$ apply jumps $N^+ = (1 - R_m)N^-$ with $R_m(x) \in [0,1]$ and $N^+ = \exp(-\alpha_{\mathrm{RT}} d_n - \beta_{\mathrm{RT}} d_n^2) N^-$ with $d_n(x) \geq 0$.

**Theorem 1** (Well-posedness and bounds). *Consider*

$$\partial_t N - D\Delta N = f(x,t,N) := k\,N\left(1 - \frac{N}{\theta}\right) - \alpha_{\mathrm{CT}} C_{\mathrm{CT}}(t)\,N \quad in\ \Omega \times (0,T) \setminus \mathcal{E},$$

*with Neumann boundary, initial data $N_0$, and the event jumps described above. Then there exists a unique weak solution $N \in L^2(0,T; H^1(\Omega)) \cap C([0,T]; L^2(\Omega))$ and for all $t \in [0,T]$,*

$$0 \leq N(\cdot, t) \leq \theta \quad a.e.\ in\ \Omega.$$

*Proof. Piecewise existence/uniqueness.* Between consecutive events the problem is a semilinear parabolic PDE with a right-hand side $f(\cdot, t, \cdot)$ that is locally Lipschitz in $N$ and has at most quadratic growth. Classical monotone operator/semigroup theory yields a unique weak solution on each open interval [see, e.g., (Evans, 2022)].

*Nonnegativity.* Let $N^- := \max\{-N, 0\}$. Testing the weak form with $N^-$ and using the Neumann boundary condition,

$$\frac{1}{2}\frac{d}{dt}\|N^-\|_{L^2}^2 + D\int_\Omega |\nabla N^-|^2\,dx = \int_\Omega f(x,t,N)\,N^-\,dx \leq C\|N^-\|_{L^2}^2,$$

with $C := k + \alpha_{\mathrm{CT}}\|C_{\mathrm{CT}}\|_{L^\infty(0,T)}$. Gronwall and $N^-(\cdot, 0) = 0$ give $N^- \equiv 0$.

*Upper bound by $\theta$.* Let $M := N - \theta$ and $M^+ := \max\{M, 0\}$. For $N \geq \theta$, we have $f(x,t,N) \leq kN(1 - N/\theta) \leq 0$. Testing with $M^+$ gives

$$\frac{1}{2}\frac{d}{dt}\|M^+\|_{L^2}^2 + D\int_\Omega |\nabla M^+|^2\,dx \leq 0.$$

Since $M^+(\cdot, 0) = 0$, we conclude $M^+ \equiv 0$, i.e., $N \leq \theta$.

*Events and gluing.* At surgery: $(1 - R_m) \in [0,1]$ maps $[0,\theta] \to [0,\theta]$. At RT: $s(x) = \exp(-\alpha_{\mathrm{RT}} d - \beta_{\mathrm{RT}} d^2) \in (0,1]$ also maps $[0,\theta] \to [0,\theta]$. Solving successively between event times and using uniqueness on each subinterval yields global uniqueness and the bound. $\square$

**Lemma 1** (Positivity of the implicit diffusion step). *Let $L_h$ be the standard finite-difference Neumann Laplacian on a uniform grid such that $-L_h$ is symmetric positive semidefinite with nonpositive off-diagonals. For $\Delta t > 0$ and $D > 0$, define $A := I - \Delta t\, D\, L_h$. Then $A$ is an $M$-matrix and $A^{-1} \geq 0$ entrywise. Consequently, the implicit diffusion substep $(I - \Delta t\, D\, L_h)\,\tilde{N} = N^n$ preserves nonnegativity: if $N^n \geq 0$ componentwise, then $\tilde{N} \geq 0$.*

*Proof.* Under the stated stencil, $-L_h$ has non-negative diagonal, non-positive off-diagonals, and is (weakly) diagonally dominant on the connected grid with reflective enforcement; its spectrum is contained in $[0, \infty)$. Thus $A$ has positive diagonal, non-positive off-diagonals, and $\sigma(A) \subset (0, \infty)$, so $A$ is an $M$-matrix with $A^{-1} \geq 0$ [See (Varga et al.) Theorem. 2.5]. $\square$

**Theorem 2** (Stability and convergence of IMEX–ETD). *Consider the Lie–Trotter IMEX step on a uniform grid:*

$$(D) \quad (I - \Delta t\, D\, L_h)\,\tilde{N} = N^n, \qquad (R) \quad N^{n+1} = \Phi_{\Delta t}(\tilde{N}),$$

*where $\Phi_{\Delta t}$ is the exact solution of $\dot{N} = aN - bN^2$ with $a := k - \alpha_{\mathrm{CT}} C_{\mathrm{CT}}(t_n)$ frozen on $[t_n, t_{n+1}]$ and $b := k/\theta$. Then:*

*(i) **Stability:** the diffusion substep is $L^2$–contractive; the full IMEX step is $L^\infty$–stable in the sense that $0 \leq N^{n+1} \leq \theta$ componentwise, and hence uniformly $L^2$–bounded.*

(ii) **Invariance:** *if* $0 \le N^n \le \theta$ *componentwise, then* $0 \le N^{n+1} \le \theta$.

(iii) **Convergence:** *the scheme is* $O(h^2 + \Delta t)$ *consistent and therefore convergent with first order in time and second order in space on* $[0, T]$ *(away from event instants; at event instants, the splitting remains first order).*

*Proof. Stability of (D).* Taking the discrete $L^2$ inner product with $\tilde{N}$ gives

$$\|\tilde{N}\|_2^2 + \Delta t\, D\, \langle -L_h \tilde{N}, \tilde{N} \rangle = \langle N^n, \tilde{N} \rangle \le \|N^n\|_2\, \|\tilde{N}\|_2,$$

which implies $\|\tilde{N}\|_2 \le \|N^n\|_2$ and monotone decay of the discrete Dirichlet form.

*Positivity/invariance.* By Lemma 1, the diffusion substep maps nonnegative data to nonnegative data. For the reaction ODE $\dot{N} = aN - bN^2$ on the interval $[0, \theta]$ we have $f(0) = 0$ and $f(\theta) = k\theta(1 - \theta/\theta) - \alpha_{\mathrm{CT}} C_{\mathrm{CT}}(t)\,\theta \le 0$. Hence by Nagumo's invariance criterion, $[0, \theta]$ is forward invariant. Equivalently, the exact reaction flow $\Phi_{\Delta t}$ maps $[0, \theta]$ into itself. Therefore if $0 \le \tilde{N} \le \theta$ componentwise, then $0 \le N^{n+1} \le \theta$.

*Convergence.* Backward Euler for diffusion is first order in time; central differences for $L_h$ are second order in space on uniform grids [ see (Thomée, 2007)]. The Lie–Trotter splitting introduces a local $O(\Delta t^2)$ error, leading to global $O(\Delta t)$ accuracy in time [see (Hundsdorfer & Verwer, 2013)]. Freezing $C_{\mathrm{CT}}$ on each step introduces only $O(\Delta t^2)$ local error when $C_{\mathrm{CT}}$ is piecewise $C^1$. Thus the global discretization error is $O(h^2 + \Delta t)$ away from events. Events are applied as exact multiplicative jumps at grid times and therefore do not degrade the first-order temporal accuracy of the overall scheme. $\square$

**Lemma 2** (Nudging preserves bounds)**.** *Let* $u \in [0, \theta]^{|\Omega|}$ *and* $u_{\mathrm{obs}} \in [0, \theta]^{|\Omega|}$ *be the model state and observed mask at a nudging time, and fix* $\alpha \in [0, 1]$. *Then* $u^{\mathrm{new}} := \alpha u + (1 - \alpha) u_{\mathrm{obs}}$ *satisfies* $0 \le u^{\mathrm{new}} \le \theta$ *componentwise.*

*Proof.* Convexity: for each voxel, $u^{\mathrm{new}}$ is a convex combination of two values in $[0, \theta]$. $\square$

**Lemma 3** (Event maps: invariance and Lipschitz)**.** *Let surgery* $J_s(N) = (1 - R) \odot N$ *with* $R \in [0, 1]^{|\Omega|}$ *and RT* $J_r(N) = S(d) \odot N$ *with* $S(d) = \exp(-\alpha_{\mathrm{RT}} d - \beta_{\mathrm{RT}} d^2) \in (0, 1)$. *Then for any* $N, M \in [0, \theta]^{|\Omega|}$,

$$0 \le J_{s/r}(N) \le \theta, \qquad \|J_{s/r}(N) - J_{s/r}(M)\|_2 \le L_{s/r} \|N - M\|_2,$$

*with* $L_s = \|1 - R\|_\infty \le 1$ *and* $L_r = \|S(d)\|_\infty \le 1$.

*Proof.* Entrywise bounds follow since $(1 - R), S(d) \in [0, 1]$. Lipschitzness is the operator norm of a diagonal map. $\square$

# E  LIMITATIONS

While the proposed SoC-DT framework shows promise, several limitations remain. First, our evaluation was performed on three synthetic toy datasets and a single real clinical dataset; broader validation across multi-institutional and heterogeneous cohorts is needed to establish clinical applicability. Second, the current implementation operates on 2D slices, which may underrepresent the full spatio-temporal complexity of tumor dynamics. Extending the framework to 3D volumetric data is an important direction for future work. Finally, additional experimentation is required to assess robustness under diverse treatment schedules and imaging protocols before translation to clinical decision support.