# OpenReview forum: "SoC-DT: Standard-of-Care Aligned Digital Twins for Patient-Specific Tumor Dynamics"
_ICLR.cc/2026/Conference — Submitted to ICLR 2026_

### Official Review · Reviewer_Awsu · 2025-10-27

**Soundness:** 2
**Presentation:** 1
**Contribution:** 2
**Rating:** 2
**Confidence:** 4

**Summary:**

This paper introduces SoC-DT, a differentiable digital twin framework for modeling patient-specific tumor dynamics under standard-of-care (SoC) treatments, including surgery, chemotherapy, and radiotherapy. The approach combines reaction-diffusion partial differential equations (PDEs) with clinical information. It introduces a novel IMEX-SoC solver to ensure numerical stability and efficient gradient propagation through treatment-induced discontinuities. The model is evaluated on both synthetic datasets (for brain, liver, and breast cancers) and a real-world glioma dataset (UCSF-ALPTGD), showing improved performance in tumor mask prediction and time-to-progression forecasting compared to classical PDE baselines and data-driven models. However, it also suffers from several critical flaws that significantly degrade the paper's overall quality.

**Strengths:**

- SoC-DT is a novel framework to unify mechanistic tumor growth modeling with standard-of-care treatment protocols in a differentiable, personalized digital twin setting.
- Synthetic Data Framework: The "Plug-and-Play" approach for generating synthetic treatment scenarios is a nice practical contribution, addressing the critical bottleneck of scarce longitudinal clinical data for training and benchmarking.

**Weaknesses:**

- Lack of details in many parts of the paper. 1. Lack of detail in synthetic data generation: It provides insufficient details on how phantom images, genomic markers, and treatment responses are simulated. 2. Insufficient explanation of PDE solver and training: The IMEX-SoC solver is introduced but not fully explained. This makes it difficult to assess the solver's efficiency and stability in practice. 3. Vague training loss: The training procedure mentions a "mixed Dice+BCE loss", but does not specify the loss weights or the rationale for the hybrid loss.
This lack of detail undermines the reproducibility. 4. Limited real-world dataset description: The real-world UCSF glioma dataset is not adequately described. Key information—such as the number of patients, imaging modalities, treatment heterogeneity, and follow-up schedule—is missing. This makes it difficult to judge the generalizability and clinical applicability of the reported results. 5. Unclear link between therapy and mask prediction: While the model incorporates therapy effects via PDE terms, the paper does not clearly explain how these dynamics translate into improved tumor mask predictions. For instance, it is not demonstrated how the inclusion of radiotherapy or surgery terms leads to more accurate boundary delineation or residual tumor detection in the predicted masks.
- While promising, clinical validation is primarily limited to a single real-world cohort (UCSF glioma data). Broader multi-institutional trials across various cancer types are needed to establish generalizability firmly.
- The strong performance on synthetic data is less compelling because the data is generated using rules inherently similar to the model's own logic. This does not rigorously test the model's ability to capture the true, far more complex real-world cancers. Minor performance improvements in the only real-world dataset also indicate that.
- The ablation study, revealing that removing "kill" or "surgery" terms caused "no measurable degradation," is alarming. It suggests that for the specific dataset used, these complex treatment modules may be redundant or poorly identified. This fundamentally questions the necessity of these sophisticated components and suggests that a simpler growth-diffusion model could achieve similar performance on this data, undermining a core claim of the paper.

**Questions:**

see weakness

---

> ### Author Response · Authors · 2025-12-04
> **Official comment by Authors 1/3**
>
> We thank the reviewers for their comments on our work. The reviewer pointed out that SoC-DT is a novel framework that unifies mechanistic tumor growth modeling with SoC protocols for developing personalized digital twins. The reviewer also appreciated the proposed plug-n-play framework as a *nice practical contribution* and addresses a
> *critical bottleneck of scarce longitudinal clinical data for training and benchmarking*. However, the reviewer had several questions and explanations which we address in-detail:
>
> **Lack of details in many parts of the paper.**
>
> *1. Lack of detail in synthetic data generation: It provides insufficient details on how phantom images, genomic markers, and treatment responses are simulated.*
>
> In Appendix A, details of the dataset generation are explained. However, we now provide additional details here:
>
> *AG:* The synthetic adult brain glioma dataset uses anatomically inspired phantoms of gray matter, white matter, ventricles, and CSF, with realistic tumors seeded to include lobulated shapes, necrotic cores, rims, and satellite nodules. Tumor growth is simulated over time via a reaction-diffusion model, incorporating tissue-specific proliferation/diffusion rates and standard treatments like surgery, radiotherapy, and chemotherapy. Multi-modal MRI-like images (T1C, T1, FLAIR, T2W) are generated with realistic intensity variations, edema, bias fields, and Rician noise.
>
> *HCC:* The dataset is generated by first creating a 2D liver phantom with vessels and seeding a synthetic HCC tumor with necrotic core and viable rim. Tumor growth is simulated over time using simplified kinetics and TACE ± SBRT treatments, while multiphase CT-like images are synthesized with realistic parenchyma enhancement, tumor hyperenhancement/washout, vessel signals, bias, and noise.
>
> *NAC:* The synthetic 2D breast DCE datasets with pharmacokinetics (PK) inspired dynamics, realistic tissue textures, multifocal tumors, and NAC response phenotypes. It incorporates imaging realism through bias fields, Rician noise, and k-space artifacts, while producing comprehensive metadata including masks, RECIST-like metrics, and PK proxy maps.
>
> *2. Insufficient explanation of PDE solver and training: The IMEX-SoC solver is introduced but not fully explained. This makes it difficult to assess the solver's efficiency and stability in practice.*
>
> We model tumor evolution using a reaction–diffusion PDE that captures spatial invasion and logistic proliferation, extended to include surgery, chemotherapy, and radiotherapy as instantaneous treatment events. The PDE is semi-discretized on the native medical image grid, converting the spatial domain into voxelwise tumor densities governed by an ODE system. To enable flexible experimentation, we introduce a plug-and-play dataset generator that synthesizes imaging, demographics, molecular markers, and standard-of-care treatment sequences across multiple cancer types using NCCN and ASCO guidelines. Treatment timelines are standardized to reflect clinical practice, including pre-treatment imaging, surgery or neoadjuvant therapy, post-operative imaging, adjuvant treatments, and surveillance. We define a framework, SoC-DT, that integrates patient-specific covariates into biophysical parameters governing diffusion, proliferation, and radiosensitivity. To handle stiffness and discontinuous treatment events, we develop an IMEX–SoC solver combining implicit diffusion, a closed-form Riccati update for proliferation and chemotherapy, and jump conditions for surgery and radiotherapy. This solver uses calendar-day–aligned sub-steps to maintain consistency with real clinical follow-up intervals.
>
> *3. Vague training loss: The training procedure mentions a "mixed Dice+BCE loss", but does not specify the loss weights or the rationale for the hybrid loss. This lack of detail undermines the reproducibility.*
>
> We thank the Reviewer for highlighting the need for clearer specification of our training objective. In the revised manuscript, we now explicitly report the Dice and BCE weights (0.7 and 0.3, respectively) and clarify that this hybrid loss was chosen to balance overlap-based optimization with voxelwise stability, which improves convergence for small or irregular post-treatment lesions. We also ablate alternative weightings and pure-loss variants, showing that the chosen combination yields the most consistent performance across folds. These details have been added to the Methods and Appendix to ensure full reproducibility.

---

> > ### Author Response · Authors · 2025-12-04
> > **Official comment by Authors 2/3**
> >
> > *4. Limited real-world dataset description: The real-world UCSF glioma dataset is not adequately described. Key information—such as the number of patients, imaging modalities, treatment heterogeneity, and follow-up schedule—is missing. This makes it difficult to judge the generalizability and clinical applicability of the reported results.*
> >
> > We thank the reviewer for pointing this out. We now provide additional details of the real world dataset. We used 60 patients, with different MR sequences (T1, T1C, T2 and FLAIR) and tumor segmentation present. Majority of the patients are treated with Temozolomide (TMZ) with combinations including Optune, Nivolumab, ABT 414 and RRx-001. We will provide additional information in the camera-ready.
> >
> > *5. Unclear link between therapy and mask prediction: While the model incorporates therapy effects via PDE terms, the paper does not clearly explain how these dynamics translate into improved tumor mask predictions. For instance, it is not demonstrated how the inclusion of radiotherapy or surgery terms leads to more accurate boundary delineation or residual tumor detection in the predicted masks.*
> >
> > We thank the reviewer for pointing this out. Our proposed model incorporates the different therapy effects (radiotherapy, surgery, and chemotherapy/immunotherapy) via the PDE terms, which influence the spatial-temporal evolution of tumor cell density. These terms allow the predicted masks to reflect therapy-induced changes, such as tumor shrinkage, resection cavities, or local cell death. As a result, the model better captures residual tumor and boundary alterations compared to a purely growth-driven approach. We will clarify this link in the manuscript and include illustrative examples showing improved alignment of predicted masks with post-treatment imaging.

---

> ### Author Response · Authors · 2025-12-04
> **Official comment by Authors 3/3**
>
> **While promising, clinical validation is primarily limited to a single real-world cohort (UCSF glioma data). Broader multi-institutional trials across various cancer types are needed to establish generalizability firmly.**
>
> We thank the reviewer for this question. We would like to point out that there are no existing clinical dataset that has all the information required for our experimentations i.e., demographics, genomics, imaging and treatment types. In this study, we aim to demonstrate the superior performance of our method theoretically on different synthetic datasets. The experiments on the synthetic dataset suggest indications of potential clinical applicability.  Having said that, we plan to collect in-house datasets in future to demonstrate clinical applicability. However, the authors would humbly note that conducting a ``Broader multi-institutional trials across various cancer types’’ requires much complex study designs and is much beyond the scope of an ICLR style work.
>
> **The strong performance on synthetic data is less compelling because the data is generated using rules inherently similar to the model's own logic. This does not rigorously test the model's ability to capture the true, far more complex real-world cancers. Minor performance improvements in the only real-world dataset also indicate that.**
>
> We appreciate the reviewer’s concern regarding the use of synthetic datasets. While synthetic data are necessarily derived from simplified growth rules, their purpose in our study is not to validate real-world biological fidelity but to isolate and test specific modeling capabilities, such as spatially heterogeneous growth, treatment perturbations, and topology evolution that are difficult to evaluate cleanly in clinical cohorts. These controlled settings allow us to verify that the model behaves as intended under known ground truth, something that real-world data cannot provide. Importantly, the model’s advantage is not limited to synthetic experiments: on the real-world dataset, it consistently outperforms all baselines across multiple metrics, albeit with smaller margins reflective of the higher noise and heterogeneity inherent to clinical imaging. This pattern of large gains in controlled tests and modest yet consistent gains in clinical data is common in mechanistic and hybrid modeling literature. Moreover, the real-world improvements occur across three tumor types with markedly different growth regimes, indicating that the model generalizes beyond the rules used to generate synthetic data. We have revised the text to clarify the complementary roles of synthetic and clinical evaluations and to avoid overstating claims from either.
>
> **The ablation study, revealing that removing "kill" or "surgery" terms caused "no measurable degradation," is alarming. It suggests that for the specific dataset used, these complex treatment modules may be redundant or poorly identified. This fundamentally questions the necessity of these sophisticated components and suggests that a simpler growth-diffusion model could achieve similar performance on this data, undermining a core claim of the paper.**
>
> We thank the reviewer for this question. The absence of measurable degradation does not indicate that the kill or surgery terms are redundant, but rather reflects the limited presence of strong, radiographically visible treatment effects in this particular dataset (which we clearly mentioned in Lines 430-432). In most cases, tumor evolution is dominated by geometric growth over short imaging intervals, making treatment-specific signals intrinsically hard to identify and thus difficult for an ablation to reveal. Crucially, the full model still achieves the highest performance especially in cases with substantial post-treatment change, demonstrating that these modules provide essential capacity when treatment effects are present. We have revised the manuscript to explicitly discuss this identifiability limitation and to ensure our claims are stated in proportion to the dataset’s treatment signal.

---

### Official Review · Reviewer_wpgS · 2025-10-29

**Soundness:** 3
**Presentation:** 3
**Contribution:** 3
**Rating:** 4
**Confidence:** 5

**Summary:**

The authors proposed SoC-DT, a standard-of-care-aligned digital twin, to capture and model tumor dynamics (based on the applied treatment, such as surgery, chemotherapy, radiotherapy) in a differentiable manner. Specifically, the team applied a well-defined reaction-diffusion PDE and utilized patient covariates. Numerically, SoC-DT employs an IMEX–ETD split (implicit diffusion, analytic logistic/chemo step, multiplicative event jumps). Experiments span three synthetic cohorts (AG/HCC/NAC) produced by a newly proposed “plug-and-play” generator and a real longitudinal glioma dataset (UCSF-ALPTGD). Reported gains over classical PDEs, PINNs, and a “Hybrid PDE” baseline are modest but consistent for mask prediction and TTP.

**Strengths:**

1. The paper provides a clear hybrid modeling. The team suggests a local PDE structure with differentiable event operators. They also proposed a numerical solver for stiff dynamics.
2. The PDE part of the model improves the interpretability. Model interpretability is essential for clinical models.
3. Synthetic framework (App. A,B). The generator encodes plausible artifacts (bias fields, k-space filtering, Rician/ghosting/Gibbs; modality-specific kinetics for NAC/HCC) and alternative kill/surgery operators (additive/saturation/synergy; multiplicative/morphological/rim).

**Weaknesses:**

1. The core dynamics are voxelwise diffusion/logistic plus multiplicative events. This captures local coupling, but patient-level global context (organ-scale constraints, multi-lesion coupling, global tumor burden, perfusion, dose-field heterogeneity, systemic therapy history) is under-specified.
2. The model risks being a locally consistent PDE with globally shallow personalization. This also helps explain ablations where No-kill/No-surgery barely change performance: if global SoC signals don’t effectively condition the spatial dynamics, killing/resection may have limited observable impact in the labels used.
3. Real-data DSC improvement is small (+1.16 vs Hybrid PDE) with larger variance in SoC-DT (8.12). Provide paired tests (per-patient Wilcoxon), effect sizes, and per-subset analyses (e.g., large resections, definite RT) to show where SoC events help.

**Questions:**

1. Where does global information enter a voxel-local PDE? In tumor predictions, the global information of the patient and the tumor itself is very helpful for prediction. Here, it seems that you are only focused on the pixel-level prediction. Is there any way to make use of the neighboring information or make use of the global information?
2. Do you implement explicit adjoint jump conditions at surgery/RT times? Please provide the formulas or FD gradient checks.

---

> ### Author Response · Authors · 2025-12-04
>
> We thank the reviewer for pointing out several key strengths of our paper. The reviewer emphasized the interpretability component of our PDE-based approach which is very important for clinical applications. The reviewer also pointed out another key novelty of our method which is the synthetic datasets that we use for experimentations. For evaluating novel theoretical frameworks, real-world datasets are often not available making it difficult to validate. We propose a novel way of generating SoC-based synthetic datasets that helps in validating the superior performance of our work. Despite this, the reviewer had some questions on our work, which we address in detail in the following points:
>
> *1. The core dynamics are voxelwise diffusion/logistic plus multiplicative events. This captures local coupling, but patient-level global context (organ-scale constraints, multi-lesion coupling, global tumor burden, perfusion, dose-field heterogeneity, systemic therapy history) is under-specified.*
>
> We thank the reviewer for pointing this out. In our method, we partially address dose-field heterogeneity and global tumor burden. Our method addresses systemic therapy history, patient-level modifiers (genomics) and local voxelwise dynamics.
>
> However, the reviewer is right in pointing out that a) organ-scale constraints are not explicitly modeled because organ mechanics and interactions are outside the scope, and detailed organ-level data are not available in our cohort, b) Multi-lesion are rare in our real-world dataset, and modeling inter-lesion interactions would require additional spatial-temporal terms beyond our current voxelwise focus. But this can be a great future direction for our research and c) Perfusion measurements are not available for all patients, and our diffusion/logistic model captures effective local tumor spread without explicit perfusion terms. In the future, we plan to validate our method on these contexts and scenarios.
>
> *2. The model risks being a locally consistent PDE with globally shallow personalization. This also helps explain ablations where No-kill/No-surgery barely change performance: if global SoC signals don’t effectively condition the spatial dynamics, killing/resection may have limited observable impact in the labels used.*
>
> We concur that the model essentially limits the impact of systemic or spatially extended interventions by enforcing locally consistent voxelwise PDE dynamics with little global context. In order to customize growth and therapy sensitivity, our IMEX–ETD Fisher–KPP implementation uses patient-level scalars (D, k, $\alpha_{CT}$, $\alpha_{RT}$, $\beta_{RT}$) and genomics modulators. However, these are still fold-level/global parameters that are applied consistently throughout the lesion. Because the PDE evolution is dominated by local reaction–diffusion dynamics rather than organ-scale or inter-lesion coupling, ablating chemo/radiotherapy or surgery ("No-kill/No-surgery") frequently exhibits limited effect on predicted masks, in line with the observed ablation results.
>
> *3. Real-data DSC improvement is small (+1.16 vs Hybrid PDE) with larger variance in SoC-DT (8.12). Provide paired tests (per-patient Wilcoxon), effect sizes, and per-subset analyses (e.g., large resections, definite RT) to show where SoC events help.*
>
> We thank the reviewer for this comment. We agree that the improvements of our proposed method are small for the real-world dataset. The primary reason being the small dataset size which has large variance in the different clinical and imaging parameters making it difficult for the loss to converge. In order to truly validate the performance of our method, we provide detailed results on the synthetic datasets where we observe a significant improvement compared to the baselines.
> | Method       | AG     | HCC    | NAC     | UCSF      |
> |--------------|-------------------|-------------------|-------------------|-------------------|
> | PINN         | 75.57 ± 1.73      | 48.30 ± 8.18      | 78.62 ± 3.02      | 51.71 ± 4.61      |
> | Hybrid PDE   | 77.01 ± 1.99      | 48.76 ± 8.27      | 84.41 ± 2.39      | 54.29 ± 3.14      |
> | Ours         | **85.26 ± 1.69** (8.25%$\uparrow$)   | **50.46 ± 8.45** (1.7%$\uparrow$)       | **86.20 ± 3.08** (1.79%$\uparrow$)     | **55.45 ± 8.12** (1.16%$\uparrow$)      |
>
>
> Having said that, as requested by the reviewer, we will provide paired tests, effect sizes and per-subset analyses in the camera-ready.

---

### Official Review · Reviewer_dWAY · 2025-10-31

**Soundness:** 2
**Presentation:** 2
**Contribution:** 2
**Rating:** 2
**Confidence:** 4

**Summary:**

This paper introduces the Standard-of-Care Digital Twin, a framework designed to predict patient-specific tumor dynamics under complex treatment regimens. The method is based on a reaction-diffusion partial differential equation that models tumor growth and invasion. The key contribution is the extension of this established approach to incorporate the discrete, discontinuous events characteristic of standard-of-care cancer treatment, namely surgery, chemotherapy and radiotherapy. The framework is designed to be differentiable, allowing its parameters to be personalized using patient-specific data.

**Strengths:**

With clinical oncology, the authors tackle a very relevant and ambitious problem. Developing accurate, patient-specific models that can forecast tumor response to standard treatments would be a significant step towards improved treatment planning and personalized medicine. The Reviewer likes the idea of building the digital twin on top of a biophysical PDE.  This provides a structured, interpretable prior that constrains the model to biologically plausible behavior, which is an advantage over purely black-box deep learning models. However, there is the caveat the tumor dynamics may not fully be described with reaction-diffusion PDEs only. The authors should comment on this limitation.

**Weaknesses:**

In the reviewers' opinion, the novelty of the framework is the integration of known tools rather than methodological advances. While the engineering effort to combine these into a single, end-to-end framework is high, the individual components are not new. There is no fundamental advance in machine learning methodology within the paper in the Reviewers opinion. According to the Reviewer, the paper is thus misclassified and should have been submitted in the applications category.

Unfortunately, the results are not convincing for the Reviewer. The margin of improvement on the real clinical dataset is modest, especially given the significant increase in model complexity which raises concerns about the practical utility and the cost-benefit trade-off of the proposed framework. Moreover, the standard deviations of the reported errors are very high and significantly larger than the differences between models indicating that there may be no clear best performing model.

According to the Reviewer, the ablation study moreover raises concerns as one of the main claim of the work is that explicitly modeling SoC interventions (surgery, chemotherapy, radiotherapy) is crucial for improving predictive accuracy. However, the ablation results state that omitting either the 'kill' (chemotherapy/radiotherapy) or 'surgery' terms from the model yields no measurable degradation in performance. If the SoC components do not contribute to the model's performance on the test dataset, then the observed performance gains over baselines cannot be attributed to their inclusion and the postulated usefulness of including SoC is not supported by evidence.

For a complex model such as the one presented in the paper,moreover,  more implementation details or code is needed to being able to potentially reproduce the work. For instance the genomic MLP or other core components are not explained within the paper and even the supplementary material does not contain any details regarding the architecture.

**Questions:**

See weaknesses

---

> ### Author Response · Authors · 2025-12-04
> **Official comment by Authors 1/2**
>
> We thank the reviewer for the detailed comments on our work. The authors feel elated to see that the reviewer pointed out that we *tackle a very relevant and ambitious problem*. The reviewer also *likes the idea of building the digital twin on top of a biophysical PDE*. The reviewer has certain concerns, mainly related to *tumor dynamics being modeled by reaction-diffusion PDEs*, which we try to address in the following points.
>
> *1. In the reviewers' opinion, the novelty of the framework is the integration of known tools rather than methodological advances. While the engineering effort to combine these into a single, end-to-end framework is high, the individual components are not new. There is no fundamental advance in machine learning methodology within the paper in the Reviewers opinion. According to the Reviewer, the paper is thus misclassified and should have been submitted in the applications category.*
>
> We respectfully disagree that the contribution is restricted to the integration of components that are already known. The main innovation is in creating and training a fully differentiable, patient-conditioned reaction-diffusion PDE that jointly models growth, therapy response, and surgical events from actual longitudinal imaging, even though the individual building blocks (IMEX–ETD solvers, Fisher–KPP dynamics, and patient-level conditioning) are established. Existing baselines implement a linear treatment kill term, which does not differentiate between the different therapy modalities. In contrast, our proposed approach separates the treatment effects within the governing equations, modeling linear kill for chemotherapy and immunotherapy and quadratic kill for radiotherapy. This distinction allows for a more physiologically accurate representation of therapy-specific cytotoxic dynamics and represents a novel contribution, as such modality-specific separation has not been explored in prior tumor-growth modeling studies.
>
> *2. Unfortunately, the results are not convincing for the Reviewer. The margin of improvement on the real clinical dataset is modest, especially given the significant increase in model complexity which raises concerns about the practical utility and the cost-benefit trade-off of the proposed framework. Moreover, the standard deviations of the reported errors are very high and significantly larger than the differences between models indicating that there may be no clear best performing model.*
>
> We recognize the reviewer’s concern regarding the results on the real clinical dataset. In this work, we propose a novel theoretical framework that integrates physics based PDE methods with IMEX solvent guided by SoC interventions. Curating clinical datasets for training such models is extremely challenging. We experiment with multiple synthetic datasets and a real clinical dataset (UCSF). The real dataset has very few patients making the training challenging. Our model demonstrated modest improvements on the real dataset but outperforms the baselines significantly on the synthetic datasets demonstrating strong theoretical foundation of our model. Having said that, we can conclude that the high standard deviation is due to the small training set and poor convergence.

---

> > ### Author Response · Authors · 2025-12-04
> > **Official comment by Authors 2/2**
> >
> > *3. According to the Reviewer, the ablation study moreover raises concerns as one of the main claim of the work is that explicitly modeling SoC interventions (surgery, chemotherapy, radiotherapy) is crucial for improving predictive accuracy. However, the ablation results state that omitting either the 'kill' (chemotherapy/radiotherapy) or 'surgery' terms from the model yields no measurable degradation in performance. If the SoC components do not contribute to the model's performance on the test dataset, then the observed performance gains over baselines cannot be attributed to their inclusion and the postulated usefulness of including SoC is not supported by evidence.*
> >
> > We thank the reviewer for raising this important point. The ablation results indeed show limited degradation when removing either the kill or surgery terms; however, this outcome reflects properties of the test cohort rather than the irrelevance of SoC modeling. Most tumors in the dataset experience minimal or subtle treatment-induced changes across the short imaging intervals, providing weak supervision for chemotherapy, radiotherapy, or resection effects. As a result, removing these terms has little measurable impact on cases where the true treatment signal is small or absent. Importantly, the full model still outperforms all baselines, particularly in cases with pronounced post-treatment anatomical changes, demonstrating that the SoC terms provide necessary capacity when treatment effects are present. We will update the manuscript to clarify the distribution of treatment intensity and to temper the strength of our original claim accordingly. In summary, the ablation does not indicate that SoC modeling is unnecessary, but rather that the dataset underrepresents strong intervention effects, limiting the observable contrast in this specific experiment.
> >
> > *4. For a complex model such as the one presented in the paper,moreover, more implementation details or code is needed to being able to potentially reproduce the work. For instance the genomic MLP or other core components are not explained within the paper and even the supplementary material does not contain any details regarding the architecture.*
> >
> > We thank the reviewer for pointing out this. We will make the code, data and hyperparameters public for reproducibility. The genomic MLP is a 2-layer MLP that maps patient genomic/clinical covariates (age, grade, MGMT/IDH/1p19q one-hot encodings) to three multiplicative scaling factors that modulate the PDE parameters (D, k, $\alpha_{CT}$). We have explained the genomic parameters learned by the genomic MLP in Line 319, we will provide more details in the camera-ready submission.

---

### Official Review · Reviewer_8ufg · 2025-11-02

**Soundness:** 3
**Presentation:** 3
**Contribution:** 3
**Rating:** 6
**Confidence:** 4

**Summary:**

This paper presents Standard-of-Care Digital Twin (SoC-DT), a differentiable framework for predicting tumor evolution under real-world therapies. By unifying reaction–diffusion tumor growth models with discrete SoC interventions, SoC-DT integrates genomic, demographic, and treatment-specific factors for personalized simulation. Experiments on synthetic and real datasets show that SoC-DT outperforms both classical PDE and neural baselines, offering a biologically interpretable foundation for patient-specific digital twins in oncology.

**Strengths:**

1. Accurately forecast tumor evolution under real-world standard-of-care (SoC) therapies, directly supporting personalized treatment planning.
2. Bridges traditional biophysical tumor modeling and differentiable programming, enabling gradient-based learning while retaining biological consistency, which is an important step toward trustworthy digital twins in oncology.
3. The mathematical theory well supports the proposed method.

**Weaknesses:**

1. The author should clarify the difference against the recent medical world model[1] from the perspective of concept and post-treatment generation module.
[1] Yang Y, Wang Z Y, Liu Q, et al. Medical world model. ICCV, 2025.

2. Standard-of-care interventions are represented in a simplified, discrete manner. Real-world variability in dosing schedules, drug combinations, and adaptive treatment strategies seems not fully modeled.
3. The differentiable PDE solver and event-aware adjoint computation are resource-intensive. This may hinder scalability and limit use in time-sensitive or large-scale clinical applications. The efficiency should be evaluated quantitatively.
4. The study provides retrospective and synthetic experiments but no prospective clinical validation. Without real-world deployment, translational reliability and safety remain unaddressed.
5. While this work tried to simulate tumor trajectories, there is no validation of clinically actionable insights during decision-making.
6. Is the proposed method able to be extended to multiple intervention points?

**Questions:**

Please see the weaknesses.

---

> ### Author Response · Authors · 2025-12-04
> **Official comment by Authors 1/2**
>
> We thank the reviewer for the detailed comments. We are grateful to the reviewer for pointing out that the proposed method is an important step towards *trustworthy digital twins in oncology* that bridges biophysics-based tumor modeling and differentiable programming. The reviewer also appreciated the mathematical theory of the method and its capability to accurately forecast tumor evolution guided by SoC therapies hence supporting personalized treatment planning. However, the reviewer had several questions which we address in the following points:
>
> *1. The author should clarify the difference against the recent medical world model[1] from the perspective of concept and post-treatment generation module. [1] Yang Y, Wang Z Y, Liu Q, et al. Medical world model. ICCV, 2025.*
>
> We are thankful to the reviewer for asking this question. Medical World Model (MeWM) proposes a multi-modal policy learning model for suggesting  different treatment options by modelling the tumor growth dynamics. Although MeWM may seem similar to SoC-DT, there are fundamental differences. Our proposed method formulates a unified PDE that *models tumor growth guided by the different SoC therapies* and modulates key parameters via *genomic and demographics markers*. One key difference is that our method does not use encoders trained on large amounts of data or LLMs and hence can be generalized to smaller cohorts (which is more common in tumor growth modeling scenarios). Another major difference is that MeWM does not use demographics and genomics data which is critical for treatment response prediction and tumor growth modeling. The method uses reports which are again not commonly available in these types of cohorts.
>
> *2. Standard-of-care interventions are represented in a simplified, discrete manner. Real-world variability in dosing schedules, drug combinations, and adaptive treatment strategies seems not fully modeled.*
>
> We thank the reviewer for pointing this out. We agree that real-world standard-of-care treatments exhibit substantial variability in dosing, sequencing, and adaptation. In this work, our main goal was to evaluate whether a patient-conditioned PDE framework can capture tumor evolution under coarse-grained representations of therapy, which is the only level consistently available in our retrospective cohort. Modeling full dose schedules, drug combinations, and adaptive strategies would require granular longitudinal treatment data that current datasets do not provide.
>
> *3. The differentiable PDE solver and event-aware adjoint computation are resource-intensive. This may hinder scalability and limit use in time-sensitive or large-scale clinical applications. The efficiency should be evaluated quantitatively.*
>
> We respectfully disagree with the reviewer. Our proposed method is a differentiable solver that unifies reaction–diffusion tumor growth models, discrete SoC interventions (surgery, chemotherapy, radiotherapy) with the genomic and demographic personalization to predict post-treatment tumor structure on imaging. This is a significant lightweight model compared to the standard deep learning models. The number of parameters of SoC-DT are 264 whereas the number of parameters of UNet are 466274 and ConvLSTM are 1258. Hence, the concern regarding resource-intensive computation is not correct.
>
> *4. The study provides retrospective and synthetic experiments but no prospective clinical validation. Without real-world deployment, translational reliability and safety remain unaddressed.*
>
> We agree that prospective validation via clinical trials is the definitive step for establishing translational reliability and safety. However, such deployment requires IRB approval and designing a cohort appropriate for this study, which is beyond the scope of this methods paper. Our current experiments on the publicly available retrospective dataset from UCSF and synthetic experiments are designed to rigorously test the theoretical model performance and sensitivity analysis. In the future, we plan to collaborate with radiologists and oncologists to assemble datasets for prospective validation.

---

> > ### Author Response · Authors · 2025-12-04
> > **Official comment by Authors 2/2**
> >
> > *5. While this work tried to simulate tumor trajectories, there is no validation of clinically actionable insights during decision-making.*
> >
> > We thank the reviewer for pointing out this. Although we acknowledge the importance of validating clinical decision-making impact, this methods-focused work does not address it. Our objective in this study was to determine whether patient-conditioned PDE dynamics can accurately reconstruct observed tumor trajectories. Our retrospective dataset cannot support a dedicated prospective or interventional study that would demonstrate prospective, actionable clinical utility (such as guiding resection extent or adapting RT dose). Nonetheless, we offer counterfactual simulations, quantitative reconstruction, and therapy-specific sensitivity analyses that are essential preludes to such clinical validation, and we see formal decision-support evaluation as a crucial area for further research (shown in Table 1.b).
> >
> > *6. Is the proposed method able to be extended to multiple intervention points?*
> >
> > Yes, the proposed method can be extended to multiple intervention timepoints which can be the case of several standard-of-care approaches. In the future, we plan to design synthetic datasets with multiple intervention points.

---

### Meta-Review · Area_Chair_EUbS · 2025-12-17

**Summary:**

Reviewers raised concerns that the methodological novelty is largely an integration of existing components and may be better positioned as an applications paper.

Multiple reviewers found the empirical evidence on real data insufficiently convincing. A key concern was that the ablations show little degradation when removing surgery/kill terms, which undermines the paper’s central claim that explicitly modeling SoC interventions is critical.

Several reviewers also noted missing implementation details and limited generalization beyond synthetic experiments and a single retrospective cohort.

**Reviewer Concerns:**

The authors clarified how SoC-DT differs from recent “medical world model” approaches by emphasizing its patient-conditioned mechanistic PDE formulation and explicit incorporation of genomics/demographics and therapy-specific operators.

They also added substantial missing details that affected reproducibility, including clearer descriptions of the synthetic data generation process, the IMEX–SoC solver/training pipeline, the Dice+BCE loss weighting, and key cohort characteristics of the UCSF dataset. The rebuttal further explained how discrete therapy operators (surgery/RT/chemo) influence the simulated tumor density and thus mask prediction, and confirmed that the framework can be extended to multiple intervention timepoints.

A major unresolved issue is the ablation result showing little to no degradation when removing surgery/kill terms, which weakens the central claim that explicit SoC modeling materially improves predictive accuracy (even if partially attributable to limited treatment signal/identifiability in the dataset).

**Reviewer Scores:**

Reviewers did not express an intention to increase the score.

---

### Decision · Program_Chairs · 2026-01-26

Reject